# UNDERSTANDING EXPONENTIAL MOVING AVERAGE: A CASE STUDY IN LINEAR REGRESSION

## ABSTRACT

Exponential moving average (EMA) has recently gained significant popularity in training modern deep learning models. However, there have been few theoretical results explaining the effectiveness of EMA. In this paper, to better understand EMA, we establish the risk bound of online SGD with EMA by performing a case study on overparameterized linear regression, which is one of the simplest over-parameterized learning tasks that shares similarities with neural networks. Our results indicate that (i) the variance error of SGD with EMA is always smaller than that of SGD without averaging, and (ii) unlike SGD with iterate averaging from the beginning, the bias error of SGD with EMA decays exponentially in every eigen-subspace of the data covariance matrix. Additionally, we develop proof techniques applicable to the analysis of a broad class of averaging schemes.

## 1 INTRODUCTION

The exponential moving average (EMA, Polyak & Juditsky 1992; Ruppert 1988) in conjunction with stochastic optimization algorithms is being extensively used in training deep learning models. By maintaining an averaged set of model parameters, EMA displays the capability to stabilize training by suppressing the noise of stochastic gradients, and it has been shown empirically that the effect of EMA is similar to that of learning rate scheduling (Sandler et al., 2023). However, this phenomenon is less studied from a theoretical perspective. Notable exceptions include a recent work by Ahn & Cutkosky (2024), which studied Adam with EMA in nonconvex optimization. However, this work is restricted to the finite-dimensional setting, which departs from the practical training of overparameterized neural networks. Block et al. (2023) revealed the variance-reducing benefit of EMA, but the bias contraction of stochastic optimization algorithms with EMA remains unknown. Meanwhile, a recent line of works (Défossez & Bach, 2015; Dieuleveut et al., 2017; Jain et al., 2018b; Berthier et al., 2020; Zou et al., 2021; Wu et al., 2022) characterized the generalization properties of SGD in overparameterized linear regression with other averaging schemes (e.g., iterate averaging from the beginning and tail averaging). In particular, Zou et al. (2021) presented an instance-dependent and dimension-free excess risk bound for SGD with iterate averaging and tail averaging. Given these results, a characterization of the generalization properties of SGD with EMA and a comparison against SGD with other averaging schemes becomes an urgent subject of study, especially in the setting of high-dimensional linear regression.

In this paper, we tackle this open problem by performing a case study of SGD with EMA in the overparameterized linear regression setting, and comparing the results with SGD without averaging, along with iterate averaging and tail averaging in Zou et al. (2021). Our contributions are summarized as follows:

- We derive the first instance-dependent excess risk bound of the linear regression model trained with SGD with EMA. We also show that the analysis is tight by presenting a lower bound that almost matches the upper bound. The excess risk bound consists of the effective bias and the effective variance, both of them further decomposed into each eigen-subspace of the data covariance matrix. Thus, the excess risk bound is only related to the eigenvalue spectrum, and is irrelevant to the ambient model dimension, making the result applicable to the overparameterized regime.

- We compare the excess risk bound of SGD with EMA against SGD without averaging as well as other averaging schemes, e.g., iterate averaging from the beginning and tail averaging, which was studied in Zou et al. (2021), summarized in Table 1. We show that (i) the effective bias of SGD with EMA decays exponentially in the number of iterations, and (ii) the effective variance of SGD with EMA is smaller than SGD without averaging, and is comparable to that of SGD with iterate averaging or tail averaging. Specifically, we observe a strong connection between EMA

Table 1: Comparison of SGD with EMA against SGD without averaging, SGD with iterate averaging from the beginning and with tail averaging. We fix the eigenvalue spectrum of the covariance matrix $\{\lambda_i\}$, the learning rate $\delta$, and the number of iterations $N$. We assume that tail averaging is performed over the last $N - s$ iterates, and $\alpha$ is the weight of the moving average in EMA. Compared with SGD without averaging which has the same exponentially decaying effective bias as EMA, SGD with EMA has a smaller variance error. SGD with either iterate averaging or tail averaging enjoys a smaller variance error than SGD without averaging, and the variance error and the effective dimension of SGD with tail averaging are identical to that of EMA when $(1 - \alpha)(N - s) = 1$. However, SGD with neither iterate averaging nor tail averaging achieves the effective bias decay rate that is exponential in $N$.

| Averaging scheme | Effective bias decay rate | Variance error in subspace of $\lambda_i$ | Eigenvalue at effective dim. |
|---|---|---|---|
| w/o averaging | Exponential in $N$ | $\mathcal{O}(\min\{\delta\lambda_i, N\delta^2\lambda_i^2\})$ | $1/(N\delta)$ |
| Iterate averaging | Polynomial in $N$ | $\mathcal{O}(\min\{1/N, N\delta^2\lambda_i^2\})$ | $1/(N\delta)$ |
| Tail averaging | Exponential in $s$ | $\mathcal{O}(\min\{1/(N-s), \delta\lambda_i, N\delta^2\lambda_i^2\})$ | $1/((N-s)\delta), 1/(N\delta)$ |
| EMA | Exponential in $N$ | $\mathcal{O}(\min\{1 - \alpha, \delta\lambda_i, N\delta^2\lambda_i^2\})$ | $(1 - \alpha)/\delta, 1/(N\delta)$ |

and tail averaging in terms of the effective variance: Suppose the tail averaging is performed over the last $N - s$ iterates in a total of $N$ iterations; if $\alpha$, the averaging parameter of EMA, satisfies $(1 - \alpha)(N - s) = 1$, then the effective variance of SGD with EMA is identical to that of SGD with tail averaging. However, the exponential decay rate of the effective bias can be achieved by SGD with tail averaging only when setting $s = \Theta(N)$ with a known training horizon $N$. This indicates that SGD with EMA has an advantage over tail averaging in the setting of unknown training budget.

- From a technical viewpoint, we identify a broad class of averaging schemes that covers all averaging methods discussed in this work. Using a standard bias-variance decomposition, we derive a crucial reformulation of the both the bias error and the variance error. Built on this reformulation, an analysis framework for all averaging schemes belonging to this class is developed in this work.

**Notations.** For a vector $\mathbf{x}$, we use $(\mathbf{x})_i$ to denote its $i$-th entry. We use $\circ$ to denote the operation of linear operators on matrices. We use $\langle \mathbf{A}, \mathbf{B} \rangle := \mathrm{tr}(\mathbf{A}\mathbf{B}^\top)$ to denote the inner product of matrices $\mathbf{A}$ and $\mathbf{B}$. For a PSD matrix $\mathbf{A}$ and a vector $\mathbf{x} \in \mathcal{H}$, define $\|\mathbf{x}\|_\mathbf{A} := \sqrt{\mathbf{x}^\top \mathbf{A} \mathbf{x}}$. For any positive integer $n$, we use $[n]$ to denote the set $\{1, 2, \ldots, n\}$. We use standard asymptotic notations $\mathcal{O}(\cdot)$, $\Omega(\cdot)$, and $\Theta(\cdot)$. We write $a \simeq b$ if there exist $c_1, c_2$ such that $c_1 a \leq b \leq c_2 a$.

## 2 RELATED WORK

**Online SGD in high-dimensional linear regression.** There is a line of works studying the excess risk bound of online SGD in the overparameterized setting using a bias-variance decomposition (Bach & Moulines, 2013; Dieuleveut & Bach, 2015; Défossez & Bach, 2015; Dieuleveut et al., 2017; Lakshminarayanan & Szepesvari, 2018; Jain et al., 2018b; Berthier et al., 2020; Zou et al., 2021; Wu et al., 2022; Lin et al., 2024). In particular, Zou et al. (2021) focused on constant-stepsize SGD with iterate averaging from the beginning or tail averaging, and derived the first instance-dependent excess risk bound of SGD in overparameterized linear regression. Wu et al. (2022) studied the last iterate risk bound of SGD with exponentially decaying stepsize, which is found to achieve a excess risk bound similar to SGD with iterate averaging. SGD with Nesterov momentum (Nesterov, 2013) and tail averaging has also been studied (Jain et al., 2018a; Varre & Flammarion, 2022; Li et al., 2023), with Li et al. (2023) obtaining an instance-dependent risk bound.

**Application of EMA.** EMA is most popular in training generative models based on GAN (Yaz et al., 2018; Karras, 2019; Kang et al., 2023), and more recently in diffusion models (Song et al., 2020b; Dhariwal & Nichol, 2021; Nichol & Dhariwal, 2021; Song et al., 2020a; Balaji et al., 2022; Karras et al., 2022; Rombach et al., 2022; Karras et al., 2024), among other applications (Block et al., 2023; Busbridge et al., 2024).

**Understanding the effect of EMA.** The favorable generalization properties of EMA in practice have been observed in several works (Tarvainen & Valpola, 2017; Izmailov et al., 2018). Through empirical experiments, Sandler et al. (2023) connected the stabilizing effect of averaging methods (e.g., EMA) with learning rate scheduling, which coincides with the finding of Wu et al. (2022). A similar theoretical result was given by Defazio (2020), but the EMA is performed on the momentum instead of the iterates.

# 3 PRELIMINARIES

## 3.1 LINEAR REGRESSION AND SGD WITH EMA

We consider the high-dimensional linear regression setting similar to Zou et al. (2021). Both the weight vectors and the data features lie within $\mathbb{R}^d$ where the ambient dimension $d$ can be arbitrarily large. The goal is to minimize the risk function

$$L(\mathbf{w}) := 1/2 \cdot \mathbb{E}_{(\mathbf{x},y)\sim\mathcal{D}}[(y - \langle \mathbf{w}, \mathbf{x}\rangle)^2],$$

where $\mathcal{D}$ is an underlying distribution of the data, $\mathbf{x} \in \mathbb{R}^d$ is the input feature vector, $y \in \mathbb{R}$ is the response, and $\mathbf{w} \in \mathcal{H}$ is the weight vector to be optimized.

We consider optimizing the objective using SGD with EMA. At iteration $t$, a random sample $(\mathbf{x}_t, y_t) \sim \mathcal{D}$ is observed, and the weight vector is updated as

$$\mathbf{w}_t = \mathbf{w}_{t-1} + \delta(y_t - \langle \mathbf{w}_{t-1}, \mathbf{x}_t\rangle)\mathbf{x}_t,$$

where $\delta > 0$ is a constant learning rate. Meanwhile, we maintain the EMA of the iterates by the following recursive formula:

$$\overline{\mathbf{w}}_0 = \mathbf{w}_0; \qquad \overline{\mathbf{w}}_t = \alpha\overline{\mathbf{w}}_{t-1} + (1-\alpha)\mathbf{w}_{t-1}, \tag{3.1}$$

where $\alpha \in (0, 1)$ is the averaging parameter. Let $N$ be the number of iterations. The final output is the $\overline{\mathbf{w}}_N$, which can be decomposed into the weighted sum of $\mathbf{w}_t$:

$$\overline{\mathbf{w}}_N = \alpha^N\mathbf{w}_0 + (1-\alpha)\sum_{t=0}^{N-1} \alpha^{N-1-t}\mathbf{w}_t. \tag{3.2}$$

## 3.2 ASSUMPTIONS

We now introduce the assumptions used in the analysis of SGD with EMA, following Zou et al. (2021); Wu et al. (2022); Li et al. (2023). The first assumption is a regularity condition that characterizes the second-order moment of the feature vector.

**Assumption 3.1** (Second-order moment). We assume that the data covariance matrix $\mathbf{H} = \mathbb{E}[\mathbf{x}\mathbf{x}^\top]$ exists. Without loss of generality, we assume that $\mathbf{H} = \mathrm{diag}(\lambda_1, \lambda_2, \dots)$ is a diagonal matrix with eigenvalues listed in descending order. We further assume that $\mathrm{tr}(\mathbf{H}) = \sum_{i=1}^{\infty} \lambda_i$ is finite. We assume that $\mathbf{H} \succ \mathbf{0}$, i.e., $L(\mathbf{w})$ admits a unique minimum $\mathbf{w}_*$.

We then present the assumptions that characterize the fourth-order moment of the data:

**Assumption 3.2** (Fourth moment condition, upper bound). We assume that the fourth moment operator $\mathcal{M} \circ \mathbf{A} = \mathbb{E}[\mathbf{x}\mathbf{x}^\top \mathbf{A}\mathbf{x}\mathbf{x}^\top]$ exists. Furthermore, there exists a scalar $\psi > 0$ such that for any PSD matrix $\mathbf{A}$, we have $\mathcal{M} \circ \mathbf{A} = \mathbb{E}[\mathbf{x}\mathbf{x}^\top \mathbf{A}\mathbf{x}\mathbf{x}^\top] \preceq \psi\,\mathrm{tr}(\mathbf{H}\mathbf{A})\mathbf{H}$.

**Assumption 3.3** (Fourth moment condition, lower bound). We assume that the fourth moment $\mathcal{M}$ exists. Furthermore, there exists a scalar $\beta > 0$ such that for any PSD matrix $\mathbf{A}$, we have $\mathcal{M} \circ \mathbf{A} - \mathbf{H}\mathbf{A}\mathbf{H} \succeq \beta\,\mathrm{tr}(\mathbf{H}\mathbf{A})\mathbf{H}$.

Assumptions about the fourth moment have been extensively used in existing works. Specifically, Zou et al. (2021) showed that if $\mathbf{H}^{-1/2}\mathbf{x}$ is $\sigma_z^2$-sub-Gaussian, then Assumption 3.2 holds with $\psi = 16\sigma_z^4$. The assumption can also be relaxed to the case where $\mathbf{A}$ is required to be PSD and commutable with $\mathbf{H}$. A special case is that the marginal distribution $\mathcal{D}|_{\mathbf{x}}$ is a Gaussian distribution. In this case, the fourth moment operator satisfies $\mathcal{M} \circ \mathbf{A} = \mathbf{H}\mathbf{A}\mathbf{H} + 2\,\mathrm{tr}(\mathbf{H}\mathbf{A})\mathbf{H}$. Note that $\mathbf{H}\mathbf{A}\mathbf{H} \preceq \mathrm{tr}(\mathbf{H}\mathbf{A})\mathbf{H}$, so we can set $\psi = 3$ in Assumption 3.2 and $\beta = 2$ in Assumption 3.3.

Finally, we present assumptions that characterize the label noise $\xi_t = y_t - \langle \mathbf{w}_*, \mathbf{x}_t\rangle$. The following assumption is a weaker condition used in the proof of the upper bound of the excess risk:

**Assumption 3.4** (Weak label noise condition). The covariance matrix of the stochastic gradient at $\mathbf{w}_*$, i.e., $\mathbf{\Sigma} := \mathbb{E}[\xi^2\mathbf{x}\mathbf{x}^\top]$ and the noise level $\sigma^2 := \|\mathbf{H}^{-\frac{1}{2}}\mathbf{\Sigma}\mathbf{H}^{-\frac{1}{2}}\|_2$ both exist and are finite.

By Assumption 3.4, we have $\mathbf{\Sigma} \preceq \sigma^2\mathbf{H}$ because $\mathbf{0} \preceq \mathbf{H}^{\frac{1}{2}}(\sigma^2\mathbf{I} - \mathbf{H}^{-\frac{1}{2}}\mathbf{\Sigma}\mathbf{H}^{-\frac{1}{2}})\mathbf{H}^{\frac{1}{2}} = \sigma^2\mathbf{H} - \mathbf{\Sigma}$.

We then present the present the stronger assumption used in the proof of the lower bound, which is referred to as the well-specified setting in the literature (Zou et al., 2021):

**Assumption 3.5** (Strong label noise condition). We assume that the label noise $\xi$ is independent of $\mathbf{x}$, and $\mathbb{E}[\xi^2] = \sigma^2$. In other words, $\mathbf{\Sigma} = \sigma^2\mathbf{H}$.

## 4 MAIN RESULTS

In this section, we present the upper and lower bounds of the excess risk, which is the difference between the risk function evaluated at the output weight vector $\overline{\mathbf{w}}_N$ and at the ground truth weight vector $\mathbf{w}_*$. Before we present the main results, we introduce the shorthand notation of sub-matrices: For any positive integers $k_1 \leq k_2$, we define $\mathbf{H}_{k_1:k_2} := \mathrm{diag}(0, \ldots, 0, \lambda_{k_1+1}, \ldots, \lambda_{k_2}, 0, \ldots)$, and $\mathbf{H}_{k_2:\infty} := \mathrm{diag}(0, \ldots, 0, \lambda_{k_2+1}, \lambda_{k_2+2}, \ldots)$. We define $\mathbf{I}_{k_1:k_2}$ and $\mathbf{I}_{k_2:\infty}$ similarly.

### 4.1 UPPER AND LOWER BOUNDS OF EXCESS RISK

**Theorem 4.1** (Upper bound). Suppose that Assumptions 3.1, 3.2 and 3.4 hold, and the hyperparameters satisfy $N(1 - \alpha) \geq 1$ and $\delta < 1/(\psi \operatorname{tr}(\mathbf{H}))$. Then the excess risk satisfies

$$\mathbb{E}[L(\overline{\mathbf{w}}_N)] - L(\mathbf{w}_*) \leq \text{EffectiveBias} + \text{EffectiveVar},$$

where the effective bias satisfies

$$\text{EffectiveBias} = \sum_{i=1}^{d} (\mathbf{w}_0 - \mathbf{w}_*)_i^2 \lambda_i \cdot b_i^2, \quad \text{where} \quad b_i := \frac{(\delta\lambda_i)\alpha^N - (1 - \alpha)(1 - \delta\lambda_i)^N}{\delta\lambda_i - (1 - \alpha)},$$

and the effective variance satisfies

$$\text{EffectiveVar} \leq \frac{1}{1 - \psi\delta\operatorname{tr}(\mathbf{H})} \left\{ \sigma^2 \underbrace{\left[ (1 - \alpha)k^* + \delta \sum_{i=k^*+1}^{k^\dagger} \lambda_i + N\delta^2 \sum_{i>k^\dagger} \lambda_i^2 \right]}_{I_1} \right.$$

$$+ \underbrace{\left[ k^*(1 - \alpha)^2 + \delta^2 \sum_{i>k^*} \lambda_i^2 \right]}_{I_2} \cdot \frac{\psi}{\delta} \cdot \underbrace{(\|\mathbf{w}_0 - \mathbf{w}_*\|_{\mathbf{I}_{0:k^\dagger}}^2 + N\delta\|\mathbf{w}_0 - \mathbf{w}_*\|_{\mathbf{H}_{k^\dagger:\infty}}^2)}_{I_3} \left. \right\},$$

where the cutoffs are defined as $k^* := \max\{i : \lambda_i \geq (1 - \alpha)/\delta\}$ and $k^\dagger := \max\{i : \lambda_i \geq 1/(N\delta)\}$.

The proof of Theorem 4.1 is given in Appendix C.1. Theorem 4.1 characterizes the first instance-dependent excess risk bound of SGD with EMA. The excess risk bound includes the effective bias and the effective variance, both decomposed into each eigen-subspace of $\mathbf{H}$. The effective bias corresponds to the convergence rate of the risk function if GD is applied instead of SGD. In the eigen-subspace corresponding to $\lambda_i$, the effective bias is $\lambda_i(\mathbf{w}_0 - \mathbf{w}_*)_i^2$, which is the initial bias error in the eigen-subspace of $\lambda_i$, multiplied by the square of the decay rate $b_i$, which will detailed in Section 4.3. The effective variance stems from the stochastic gradient, including the randomness of both $\mathbf{x}_t$ and $y_t$. We will discuss key elements of the effective variance in Section 4.2. Specifically, $I_1$, $I_2$ and $I_3$ can be seen as the sum of the factors from each eigen-subspace (see (4.1), (4.2), and (4.3)), with the eigen-subspaces classified into three regimes: $i \leq k^*$, $k^* < i \leq k^\dagger$, and $i > k^\dagger$.

We also obtain the lower bound of the excess risk of SGD with EMA:

**Theorem 4.2** (Lower bound). Suppose that Assumptions 3.1, 3.3 and 3.5 hold, and the hyperparameters satisfy $\delta \leq 1/\lambda_1$, $\alpha^{N-1} \leq 1/N$, and $N \geq 2$. The excess risk then satisfies

$$\mathbb{E}[L(\overline{\mathbf{w}}_N)] - L(\mathbf{w}_*) = (\text{EffectiveBias} + \text{EffectiveVar})/2,$$

where the effective bias is identical to that in Theorem 4.1, and the effective variance satisfies

$$\text{EffectiveVar} \gtrsim (\beta\|\mathbf{w}_0 - \mathbf{w}_*\|_{\mathbf{H}_{k^\dagger:\infty}}^2 + \sigma^2) \cdot \left[ (1 - \alpha)k^* + \delta \sum_{i=k^*+1}^{k^\dagger} \lambda_i + N\delta^2 \sum_{i>k^\dagger} \lambda_i^2 \right].$$

The lower bound is matching with the upper bound except for the second term of the effective variance, which will be discussed in Subsection 4.2. Although Theorem 4.2 requires a stronger condition about $N$ and $\alpha$, it is still a mild condition in practice because $\alpha^{N-1}$, which is the weight of $\mathbf{w}_0$ in (3.2), should be smaller than the average weight $1/N$.

### 4.2 DISCUSSION OF VARIANCE ERROR

**Effective dimensions.** The cutoffs $k^*$ and $k^\dagger$ are referred to as *effective dimensions*, which can be significantly smaller than the real model dimension $d$, especially when the eigenvalue spectrum decays fast. Similar quantities also appear in previous works analyzing high-dimensional linear regression (Zou et al., 2021; Wu et al., 2022; Li et al., 2023), and the double effective dimensions $k^*$ and $k^\dagger$ for SGD with EMA is very similar to that of SGD with tail averaging (Zou et al., 2021). We will draw more connections between SGD with EMA and SGD with tail averaging in Section 5.

We now discuss the effective variance error with the help of the effective dimensions. Both the upper bound and the lower bound of the effective variance contain two terms:

- The first term, which is determined by $I_1$ in Theorem 4.1, stems from the label noise, and is referred to as the *(real) variance error*. The upper bound and the lower bound are matching for this term up to constant factors. In each eigen-subspace in the regime $i \leq k^*$ corresponding to eigenvalue $\lambda_i \geq (1 - \alpha)/\delta$, the variance error is $\Theta(1 - \alpha)$. In eigen-subspaces with $i > k^*$, the variance error $\min\{\delta\lambda_i, N\delta^2\lambda_i^2\}$ decays with the decay of the eigenvalue $\lambda_i$. Under the source condition $\lambda_i \simeq i^{-a}$ (Caponnetto & De Vito, 2007), the total variance error is $\Theta(\delta^{1/a}(1-\alpha)^{1-1/a})$. Therefore, even when $d \to \infty$, the variance error remains finite as long as $a > 1$.
- The second term, which is the product of $I_2$ and $I_3$ in Theorem 4.1, comes from the randomness of the feature vector, and is thus nonzero even if there is no label noise. The upper and lower bounds are not matching for this term due to the additional term $\|\mathbf{w}_0 - \mathbf{w}_*\|^2_{\mathbf{I}_{0:k^\dagger}}$ in the upper bound, which is similar to the case of SGD with tail averaging (Zou et al., 2021). We conjecture that finer analysis can bridge the gap. $I_2$ and $I_3$ are finite under the source condition, similar to $I_1$.

We then discuss the influence of hyperparameters $\delta$, $\alpha$, and $N$ on the effective variance bound. The following equations that decompose $I_1$, $I_2$, and $I_3$ into eigen-subspaces will be helpful in the discussion about the effect of hyperparameters on the effective variance:

$$I_1 = \sum\nolimits_{i=1}^{d} \min\{1 - \alpha, \delta\lambda_i, N\delta^2\lambda_i^2\}; \tag{4.1}$$

$$I_2 = \sum\nolimits_{i=1}^{d} (\min\{1 - \alpha, \delta\lambda_i\})^2; \tag{4.2}$$

$$I_3 = \sum\nolimits_{i=1}^{d} \lambda_i(\mathbf{w}_0 - \mathbf{w}_*)_i^2 \min\{1, N\delta\lambda_i\}. \tag{4.3}$$

**Learning rate $\delta$.** In the upper bound of the excess risk (Theorem 4.1), we require that $\delta < 1/(\psi \operatorname{tr}(\mathbf{H}))$ similar to Zou et al. (2021), to ensure that $(1 - \psi\delta \operatorname{tr}(\mathbf{H}))^{-1}$ is positive. Larger learning rates may cause the effect of the fourth moment to accumulate and diverge.

**Number of iterations $N$.** Due to (4.3) and (4.1), the effective variance increases as $N$ increases. Furthermore, as $N$ goes to infinity, $k^\dagger$ also goes to infinity, while $k^*$ remains unchanged.

**Averaging parameter $\alpha$.** Due to (4.2) and (4.1), the effective variance decreases as $\alpha$ increases. However, choosing $\alpha$ very close to 1 does not truly benefit the learning process because the reduced variance error stems partly from the large weight of $\mathbf{w}_0$ (which has no randomness) in (3.2). We will further elaborate this point in the next subsection.

### 4.3 Decay Rate of Bias Error

We then study the quantity $b_i$ in Theorems 4.1 and 4.2, which is the decay rate of the effective bias in the eigen-subspace of $\lambda_i$. We first note that

$$b_i = (1 - \delta\lambda_i)^N + (\delta\lambda_i) \sum\nolimits_{t=0}^{N-1} \alpha^t(1 - \delta\lambda_i)^{N-1-t},$$

so the smaller $\alpha$ is, the faster $b_i$ decays. Together with the analysis of the effective variance in Subsection 4.2, we conclude that there exists a bias-variance trade-off concerning the choice of $\alpha$: Larger $\alpha$ brings about smaller effective variance, but makes the effective bias decay slower.

The following proposition presents a finer characterization of the decay rate $b_i$:

**Proposition 4.3.** In the eigen-subspace of $\lambda_i$ for any $i \in [d]$, the exponential decay rate $b_i$ satisfies

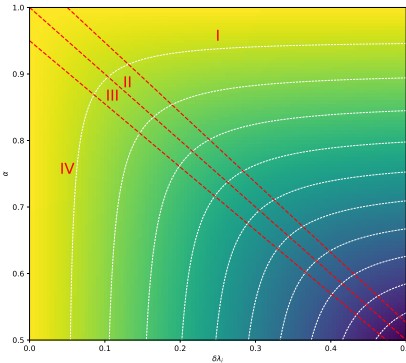

Figure 1: Heatmap and contours of $b_i$ (in log scale) with respect to $\alpha$ and $\delta\lambda_i$. We select $N = 20$ to make the regimes clear.

$$b_i \simeq \begin{cases} (\delta\lambda_i)\alpha^N/(\delta\lambda_i - (1 - \alpha)) & \text{Regime I:} \quad (1 - \delta\lambda_i)/\alpha \in (0, (N-1)/N]; \\ (1 - \alpha)N\alpha^{N-1} & \text{Regime II:} \quad (1 - \delta\lambda_i)/\alpha \in ((N-1)/N, 1]; \\ \delta\lambda_i N(1 - \delta\lambda_i)^{N-1} & \text{Regime III:} \quad (1 - \delta\lambda_i)/\alpha \in (1, N/(N-1)]; \\ (1 - \alpha)(1 - \delta\lambda_i)^N/((1 - \alpha) - \delta\lambda_i) & \text{Regime IV:} \quad (1 - \delta\lambda_i)/\alpha \in (N/(N-1), \infty). \end{cases}$$

Proposition 4.3 implies that (i) the effective bias decays exponentially in $N$ within every eigen-subspace of $\mathbf{H}$; (ii) the decay rate of the effective bias has a phase transition at the eigen-subspace

Table 2: Interpretation of four regimes of eigen-subspaces.

| Regime | Range of $(1 - \delta\lambda_i)/\alpha$ | Interpretation | Dependence of $b_i$ on $N$ |
|--------|------------------------------------------|----------------|----------------------------|
| I | $(0, (N-1)/N]$ | $\delta\lambda_i$ significantly larger than $1 - \alpha$ | $\alpha^N$ |
| II | $((N-1)/N, 1)$ | $\delta\lambda_i$ slightly larger than $1 - \alpha$ | $N\alpha^{N-1}$ |
| III | $(1, N/(N-1)]$ | $\delta\lambda_i$ slightly smaller than $1 - \alpha$ | $N(1 - \delta\lambda_i)^{N-1}$ |
| IV | $(N/(N-1), \infty)$ | $\delta\lambda_i$ significantly smaller than $1 - \alpha$ | $(1 - \delta\lambda_i)^N$ |

corresponding to $\lambda_{k^*}$: The decay rate is $\alpha^{2N}$ in the eigen-subspace of large eigenvalues, and is $(1 - \delta\lambda_i)^{2N}$ in the eigen-subspace of small eigenvalues, and (iii) when $1 - \delta\lambda_i$ is close to $\alpha$, the decay rate of the effective bias contains additional factors polynomial in $N$. Figure 1 presents an illustration of Proposition 4.3 with the four regimes of the eigen-subspaces: In Regime I, i.e., when $\lambda_i$ is large, $b_i$ is mainly affected by $\alpha$ due to the exponential term $\alpha^N$, while in Regime IV, i.e., when $\delta\lambda_i$ is small, $b_i$ is mainly affected by $\delta\lambda_i$, due to the exponential term $(1 - \delta\lambda_i)^N$. Regimes II and III stand for the case where $1 - \delta\lambda_i$ is close to $\alpha$. We also summarize the four regimes in Table 2.

**Heuristics on choice of $\alpha$.** We now discuss the optimal choice of $\alpha$ under the source condition $\lambda_i \simeq i^{-a}$ and $\lambda_i(\mathbf{w}_0 - \mathbf{w}_*)_i^2 = i^{-b}$. To simplify discussions, we only consider the real variance error, i.e., the term $I_1$ in Theorem 4.1, in the effective variance, which is already shown to be $\Theta(\delta^{1/a}(1-\alpha)^{1-1/a})$. For the effective bias error, we approximate $b_i$ with $\alpha^N$ for $i \le k^*$, then

$$\sum_{i \le k^*} \lambda_i(\mathbf{w}_0 - \mathbf{w}_*)_i^2 \cdot b_i^2 \simeq \sum_{i \le k^*} i^{-b}\alpha^{2N} \simeq \alpha^{2N}.$$

For $i > k^*$, we approximate $b_i$ with $(1 - \delta\lambda_i)^N$, then

$$\sum_{i > k^*} \lambda_i(\mathbf{w}_0 - \mathbf{w}_*)_i^2 \cdot b_i^2 \simeq \sum_{i > k^*} i^{-b}(1 - \delta i^{-a})^{2N} \approx \int_{k^*}^{\infty} x^{-b}(1 - \delta x^{-a})^{2N}\,\mathrm{d}x$$

$$= \delta^{-b/a}\int_0^{1-\alpha} y^{b/a}(1-y)^{2N}\,\mathrm{d}y \le \delta^{-b/a}B(b/a + 1, 2N + 1),$$

where $B(\cdot, \cdot)$ is the Beta function. We observe that the upper bound of $\sum_{i > k^*} \lambda_i(\mathbf{w}_0 - \mathbf{w}_*)_i^2 \cdot b_i^2$ does not depend on $\alpha$. Therefore, in order to find $\alpha$ to minimize the excess risk, it suffices to find $\alpha$ to minimize the sum of the variance error and the effective bias for $i \le k^*$, i.e., $\delta^{1/a}(1-\alpha)^{1-1/a} + \alpha^{2N}$.

## 5 COMPARING EMA WITH OTHER AVERAGING SCHEMES

In this section, we compare the excess risk of SGD with EMA against SGD without averaging and other averaging schemes, including iterate averaging from the beginning and tail averaging. Similar to EMA, the excess risk of all averaging schemes of interest can be decomposed into effective bias and effective variance (Zou et al., 2021). For each averaging scheme, we focus on its comparison with EMA in terms of effective variance (including the effective dimension) and the decay rate of the effective bias, i.e., $b_i$.

**Comparison with SGD without averaging.** SGD without averaging is equivalent to EMA with $\alpha = 0$. Specifically, the effective dimension $k^*$ becomes 0, and the decay rate of the effective bias is $b_i^{\text{w/o}} = (1 - \delta\lambda_i)^{N-1}$. Based on the discussion about the impact of $\alpha$ on the excess risk bound in Subsections 4.2 and 4.3, we conclude that *SGD with EMA has a smaller effective variance, but its effective bias decays slower than that of SGD without averaging.*

**Comparison with iterate averaging.** Zou et al. (2021) studied SGD with iterate averaging, which is defined as $\overline{\mathbf{w}}_N^{\text{IA}} := N^{-1}\sum_{t=0}^{N-1} \mathbf{w}_t$. The variance error of SGD with iterate averaging is

$$\Theta\left(\sigma^2 \sum_{i=1}^d \min\{1/N, N\delta^2\lambda_i^2\}\right).$$

If $N$ is not too large, i.e., $N\alpha^{N-1} = \Theta(1)$, the difference between $1/N$ and $1 - \alpha$ is only $\text{polylog}(N)$. In this case, *SGD with EMA achieves a variance error similar to that of SGD with iterate averaging.* Due to the gap between the upper and lower bounds of SGD with EMA, we leave the comparison of the remaining part of the effective variance for future work. The decay rate of effective bias of SGD with iterate averaging is $b_i^{\text{IA}} = (1 - (1 - \delta\lambda_i)^N)/(N\delta\lambda_i) = \Theta(\min\{1/(N\delta\lambda_i), 1\})$.

Therefore, *SGD with EMA enjoys the advantage of exponentially decaying effective variance compared with SGD with iterate averaging.*

**Comparison with tail averaging.** Zou et al. (2021) also studied SGD with tail averaging, where averaging is only performed for the last $N - s$ iterates, i.e., $\overline{\mathbf{w}}_{s:N}^{\mathrm{TA}} := (N - s)^{-1} \sum_{t=s}^{N-1} \mathbf{w}_t$. Similar to the case in Subsection 4.1, the upper and lower bounds of the excess risk of SGD with tail averaging are not matching in Zou et al. (2021), so we focus on the comparison of the effective dimension and the real variance error in the effective variance. According to Zou et al. (2021), the effective dimensions of SGD with tail averaging are $k_{\mathrm{TA}}^* = \max\{i : \lambda_i \geq 1/((N - s)\delta)\}$ and $k_{\mathrm{TA}}^\dagger = \max\{i : \lambda_i \geq 1/(N\delta)\}$. We thus observe that $k_{\mathrm{TA}}^\dagger$ *is exactly the same as* $k^\dagger$ *in SGD with EMA, and* $k_{\mathrm{TA}}^* = k^*$ *under the condition* $(1 - \alpha)(N - s) = 1$. Furthermore, the real variance of SGD with tail averaging is

$$\mathrm{Variance} = \Theta\big(\sigma^2 \sum\nolimits_{i=1}^{d} \min\big\{1/(N - s), \delta\lambda_i, N\delta^2\lambda_i^2\big\}\big),$$

which also *matches that of SGD with EMA if* $(1 - \alpha)(N - s) = 1$. For the decay rate of the effective bias, we have $b_i^{\mathrm{TA}} = ((1 - \delta\lambda_i)^s - (1 - \delta\lambda_i)^N)/((N - s)\delta\lambda_i)$. We then compare $b_i$ with $b_i^{\mathrm{TA}}$ under the condition $(1 - \alpha)(N - s) = 1$. When $\alpha \geq 1/2$ (which is a mild condition in practice), we have $\log \alpha \geq (\alpha - 1)/2$, and $1/\sqrt{e} = e^{(\alpha-1)(N-s)/2} \leq e^{(N-s)\log\alpha} = \alpha^{N-s}$. We thus have

$$b_i = (1 - \alpha) \sum\nolimits_{t=s}^{N-1} \alpha^{N-1-t}(1 - \delta\lambda_i)^t + \alpha^{N-s}\frac{(\delta\lambda_i)\alpha^s - (1 - \alpha)(1 - \delta\lambda_i)^s}{\delta\lambda_i - (1 - \alpha)}$$

$$\geq \frac{1 - \alpha}{\sqrt{e}} \sum\nolimits_{t=s}^{N-1} (1 - \delta\lambda_i)^t = \frac{(1 - \delta\lambda_i)^s - (1 - \delta\lambda_i)^N}{\sqrt{e}(N - s)\delta\lambda_i},$$

where the inequality holds due to a dropped positive term and $\alpha^{N-s} \geq 1/\sqrt{e}$. Therefore, the exponential decay rate of SGD with EMA $b_i$ is $\Omega(b_i^{\mathrm{TA}})$. However, $b_i$ is exponential in $N$ while $b_i^{\mathrm{TA}}$ is exponential only in $s$, which means that *SGD with EMA has the advantage that the effective bias in every eigen-subspace decays exponentially fast in $N$ compared with polynomial decay in $N$ for SGD with tail averaging if $s$ is fixed before training.*

## 6 EXTENSION TO MINI-BATCH SGD

We now extend our analysis of SGD with EMA to mini-batch SGD. Let $B$ be the batch size, and $\{(\mathbf{x}_{t,i}, y_{t,i})\}_{i=1}^{B}$ be the mini-batch sampled from the distribution $\mathcal{D}$ at iteration $t$. An iterate of mini-batch SGD is

$$\mathbf{w}_t^{\mathrm{MB}} = \mathbf{w}_{t-1}^{\mathrm{MB}} + \delta/B \cdot \sum\nolimits_{i=1}^{B} (y_{i,t} - \langle \mathbf{w}_{t-1}^{\mathrm{MB}}, \mathbf{x}_{t,i} \rangle)\mathbf{x}_{t,i}.$$

We then consider the excess risk of the exponential moving average of the mini-batch SGD iterates, defined as

$$\overline{\mathbf{w}}_N^{\mathrm{MB}} = \alpha^N \mathbf{w}_0^{\mathrm{MB}} + (1 - \alpha) \sum\nolimits_{t=0}^{N-1} \alpha^{N-1-t}\mathbf{w}_t^{\mathrm{MB}}.$$

**Theorem 6.1.** Suppose that Assumptions 3.1, 3.2, and 3.4 hold, and the learning rate satisfies $\delta < \min\{B/(2\psi \operatorname{tr}(\mathbf{H})), 1/\|\mathbf{H}\|_2\}$. Then the excess risk of mini-batch SGD satisfies

$$\mathbb{E}[L(\overline{\mathbf{w}}_N)] - L(\mathbf{w}_*) \leq \textcolor{red}{\mathrm{EffectiveBias}} + \textcolor{red}{\mathrm{EffectiveVar}},$$

where the effective bias is identical to that in Theorem 4.1, and the excess variance satisfies

$$\textcolor{blue}{\mathrm{EffectiveVar}} \leq 2\sigma^2/B \cdot \Big[(1 - \alpha)k^* + \delta \sum\nolimits_{i=k^*+1}^{k^\dagger} \lambda_i + N\delta^2 \sum\nolimits_{i>k^\dagger} \lambda_i^2\Big]$$

$$+ 2\psi/(\delta B) \cdot \big(k^*(1 - \alpha)^2 + \delta^2 \sum\nolimits_{i>k^*} \lambda_i^2\big) \cdot \big(2\psi(\|\mathbf{w}_0 - \mathbf{w}_*\|_{\mathbf{I}_{0:k^\dagger}}^2 + N\delta\|\mathbf{w}_0 - \mathbf{w}_*\|_{\mathbf{H}_{k^\dagger:\infty}}^2)\big).$$

A lower bound corresponding to Theorem 6.1 can be proved similar to Theorem 4.2.

Based on Theorem 6.1, we aim to derive the critical batch size (Zhang et al., 2024), which is the batch size that causes a phase transition on the excess risk bound. Since the effective variance decays exponentially in $N$, we present the following corollary for only the effective variance:

**Corollary 6.2.** Suppose the eigenvalue spectrum satisfies $\lambda_i = i^{-a}$, and the initialization satisfies $\lambda_i(\mathbf{w}_0 - \mathbf{w}_*)_i^2 = i^{-b}$ where $b < a + 1$. Let $M$ be the number of examples. Then under the same assumptions as Theorem 6.1, we have

$$\textcolor{blue}{\mathrm{EffectiveVar}} = \Theta(B^{-1}\delta^{1/a}(1 - \alpha)^{1-1/a}) + \Theta(B^{-1}\delta^{(2-b)/a}(1 - \alpha)^{2-1/a}N^{1-(b-1)/a}).$$

The assumption of the eigenvalue spectrum and the initialization is referred to as the source condition (Caponnetto & De Vito, 2007; Zhang et al., 2024). The assumption of $b < a + 1$ ensures that upper bound and the lower bounds are matching. If we further let $N = M/B$ where $M$ is the total number of samples, then the critical batch size is $B^* = \mathcal{O}(M\delta^{\frac{1-b}{a-b+1}}(1-\alpha)^{\frac{a}{a-b+1}})$. We observe that the critical batch size of SGD with EMA is sharply different from SGD with iterate averaging in Zhang et al. (2024). This is because the critical batch size is determined by both the effective bias and the effective variance for SGD with iterate averaging due to the effective bias that decays only polynomially in $N$. However, the effective bias of SGD with EMA decays exponentially in $N$, making it negligible in the analysis of the critical batch size.

# 7 OVERVIEW OF PROOF TECHNIQUES

In this section, we present the proof technique that is not only used in our analysis of EMA, but also applicable to a class of averaging schemes.

We first introduce the class of averaging schemes that covers EMA and iterate averaging, among others. In (3.1), instead of using a uniform $\alpha$ in all iterates, we allow the averaging parameter to depend on $t$, i.e.,

$$\overline{\mathbf{w}}_0 = \mathbf{w}_0; \qquad \overline{\mathbf{w}}_t = \alpha_{t-1}\overline{\mathbf{w}}_{t-1} + (1 - \alpha_{t-1})\mathbf{w}_{t-1}.$$

where $\alpha_t \in [0, 1]$ is the time-dependent averaging parameter. The final output can be written as

$$\overline{\mathbf{w}}_N = \beta_0 \mathbf{w}_0 + \sum_{t=0}^{N-1} (\beta_{t+1} - \beta_t)\mathbf{w}_t,$$

where $\beta_t$ is defined as $\beta_t = \prod_{k=t}^{N-1} \alpha_t$. Most averaging schemes belong to this class, e.g.,

- EMA: $\alpha_t = \alpha$, and $\beta_t = \alpha^{N-t}$.
- SGD without averaging: $\alpha_t = 0$, $\beta_N = 1$, and $\beta_t = 0$ for all $t = 0, \ldots, N - 1$.
- Iterate averaging: $\alpha_t = t/(t + 1)$, and $\beta_t = t/N$.
- Tail averaging:

$$\alpha_t = \begin{cases} 0 & t < s, \\ \frac{t-s}{t-s+1} & t \geq s; \end{cases}, \qquad \beta_t = \begin{cases} 0 & t < s, \\ \frac{t-s}{N-s} & t \geq s. \end{cases}$$

We now define several notations following Zou et al. (2021). We first define the centered SGD iterate as $\boldsymbol{\eta}_t = \mathbf{w}_t - \mathbf{w}_*$, and the EMA of the centered SGD iterates is $\overline{\boldsymbol{\eta}}_N = \overline{\mathbf{w}}_N - \mathbf{w}_*$. We define the centered bias and variance vectors recursively as

$$\boldsymbol{\eta}_0^{\text{bias}} = \boldsymbol{\eta}_0, \quad \boldsymbol{\eta}_t^{\text{bias}} = (\mathbf{I} - \delta\mathbf{x}_t\mathbf{x}_t^\top)\boldsymbol{\eta}_{t-1}^{\text{bias}};$$

$$\boldsymbol{\eta}_0^{\text{var}} = \mathbf{0}, \quad \boldsymbol{\eta}_t^{\text{var}} = (\mathbf{I} - \delta\mathbf{x}_t\mathbf{x}_t^\top)\boldsymbol{\eta}_{t-1}^{\text{var}} + \delta\xi_t\mathbf{x}_t.$$

We can define the EMA of the centered vectors $\overline{\boldsymbol{\eta}}_N$, $\overline{\boldsymbol{\eta}}_N^{\text{bias}}$, and $\overline{\boldsymbol{\eta}}_N^{\text{var}}$ similar to the definition of $\overline{\mathbf{w}}_N$ in (3.2). Following previous works (Défossez & Bach, 2015; Dieuleveut et al., 2017; Jain et al., 2018b; Berthier et al., 2020; Zou et al., 2021; Wu et al., 2022; Lin et al., 2024; Li et al., 2023), under Assumption 3.4, the excess risk can be decomposed as (See Lemma C.1 for details) $\mathbb{E}[L(\overline{\mathbf{w}}_N)] - L(\mathbf{w}_*) \leq \text{bias} + \text{var}$, where the bias and variance errors are defined as

$$\text{bias} = \langle \mathbf{H}, \mathbb{E}[\overline{\boldsymbol{\eta}}_N^{\text{bias}} \otimes \overline{\boldsymbol{\eta}}_N^{\text{bias}}]\rangle, \quad \text{var} = \langle \mathbf{H}, \mathbb{E}[\overline{\boldsymbol{\eta}}_N^{\text{var}} \otimes \overline{\boldsymbol{\eta}}_N^{\text{var}}]\rangle. \tag{7.1}$$

Since $\overline{\boldsymbol{\eta}}_N^{\text{bias}}$ and $\overline{\boldsymbol{\eta}}_N^{\text{var}}$ are the weighted sums of $\boldsymbol{\eta}_t^{\text{bias}}$ and $\boldsymbol{\eta}_t^{\text{var}}$, respectively, in order to bound bias and var which depends on the covariance matrix of $\overline{\boldsymbol{\eta}}_N^{\text{bias}}$ and $\overline{\boldsymbol{\eta}}_N^{\text{var}}$, it suffices to (i) study terms of the form $\mathbb{E}[\boldsymbol{\eta}_t^{\text{bias}} \otimes \boldsymbol{\eta}_k^{\text{bias}}]$ and $\mathbb{E}[\boldsymbol{\eta}_t^{\text{var}} \otimes \boldsymbol{\eta}_k^{\text{var}}]$, and (ii) represent the bias and variance errors in a tractable form. For Step (i), following Zou et al. (2021), we define the covariance matrices as $\mathbf{B}_t = \mathbb{E}[\boldsymbol{\eta}_t^{\text{bias}} \otimes \boldsymbol{\eta}_t^{\text{bias}}]$ and $\mathbf{C}_t = \mathbb{E}[\boldsymbol{\eta}_t^{\text{var}} \otimes \boldsymbol{\eta}_t^{\text{var}}]$. With these definitions, for $k \geq t$, we have $\mathbb{E}[\boldsymbol{\eta}_t^{\text{bias}} \otimes \boldsymbol{\eta}_k^{\text{bias}}] = \mathbf{B}_t(\mathbf{I} - \delta\mathbf{H})^{k-t}$ and $\mathbb{E}[\boldsymbol{\eta}_t^{\text{var}} \otimes \boldsymbol{\eta}_k^{\text{var}}] = \mathbf{C}_t(\mathbf{I} - \delta\mathbf{H})^{k-t}$. We are now ready to represent $\mathbb{E}[\overline{\boldsymbol{\eta}}_N^{\text{bias}} \otimes \overline{\boldsymbol{\eta}}_N^{\text{bias}}]$ using $\mathbf{B}_t$:

$$\mathbb{E}[\overline{\boldsymbol{\eta}}_N^{\text{bias}} \otimes \overline{\boldsymbol{\eta}}_N^{\text{bias}}] = \beta_0^2 \mathbf{B}_0 + \sum_{t=0}^{N-1} \beta_0(\beta_{t+1} - \beta_t)[(\mathbf{I} - \delta\mathbf{H})^t \mathbf{B}_0 + \mathbf{B}_0(\mathbf{I} - \delta\mathbf{H})^t]$$

$$+ \sum_{t=0}^{N-1} (\beta_{t+1} - \beta_t) \Bigg[ (\beta_{t+1} - \beta_t) \mathbf{B}_t + \sum_{k=t+1}^{N-1} (\beta_{k+1} - \beta_k)[(\mathbf{I} - \delta\mathbf{H})^{k-t}\mathbf{B}_t + \mathbf{B}_t(\mathbf{I} - \delta\mathbf{H})^{k-t}] \Bigg].$$

$$(7.2)$$

For Step (ii), the analysis in Zou et al. (2021); Wu et al. (2022) that adds the terms $\mathbf{B}_t$ and transforms (7.2) into a "triangular" sum does not work due to the *inhomogeneous* $\beta_t - \beta_{t+1}$. To tackle this issue, we make the critical observation that

$$(\beta_{t+1} - \beta_t) \Bigg[ (\beta_{t+1} - \beta_t) \mathbf{B}_t + \sum_{k=t+1}^{N-1} (\beta_{k+1} - \beta_k)[(\mathbf{I} - \delta\mathbf{H})^{k-t}\mathbf{B}_t + \mathbf{B}_t(\mathbf{I} - \delta\mathbf{H})^{k-t}] \Bigg]$$

$$= \Bigg[ \sum_{k=t}^{N-1} (\beta_{k+1} - \beta_k)(\mathbf{I} - \delta\mathbf{H})^{k-t} \Bigg] \cdot \mathbf{B}_t \cdot \Bigg[ \sum_{k=t}^{N-1} (\beta_{k+1} - \beta_k)(\mathbf{I} - \delta\mathbf{H})^{k-t} \Bigg]$$

$$- \Bigg[ \sum_{k=t+1}^{N-1} (\beta_{k+1} - \beta_k)(\mathbf{I} - \delta\mathbf{H})^{k-t-1} \Bigg] \cdot (\widetilde{\mathcal{B}} \circ \mathbf{B}_t) \cdot \Bigg[ \sum_{k=t+1}^{N-1} (\beta_{k+1} - \beta_k)(\mathbf{I} - \delta\mathbf{H})^{k-t-1} \Bigg],$$

where the matrix operator $\widetilde{\mathcal{B}}$ is defined as $\widetilde{\mathcal{B}} = (\mathbf{I} - \delta\mathbf{H}) \otimes (\mathbf{I} - \delta\mathbf{H})$. Similar properties were first used in Li et al. (2023) to study the generalization of SGD with Nesterov momentum. Using this property, by applying the telescope sum, (7.2) can be reformulated as

$$\mathbb{E}[\overline{\boldsymbol{\eta}}_N^{\text{bias}} \otimes \overline{\boldsymbol{\eta}}_N^{\text{bias}}] = \Bigg[ \beta_0 \mathbf{I} + \sum_{k=0}^{N-1} (\beta_{k+1} - \beta_k)(\mathbf{I} - \delta\mathbf{H})^k \Bigg] \mathbf{B}_0 \Bigg[ \beta_0 \mathbf{I} + \sum_{k=0}^{N-1} (\beta_{k+1} - \beta_k)(\mathbf{I} - \delta\mathbf{H})^k \Bigg]$$

$$+ \sum_{t=1}^{N-1} \Bigg[ \sum_{k=t}^{N-1} (\beta_{k+1} - \beta_k)(\mathbf{I} - \delta\mathbf{H})^{k-t} \Bigg] (\mathbf{B}_t - \widetilde{\mathcal{B}} \circ \mathbf{B}_{t-1}) \Bigg[ \sum_{k=t}^{N-1} (\beta_{k+1} - \beta_k)(\mathbf{I} - \delta\mathbf{H})^{k-t} \Bigg],$$

$$(7.3)$$

where the first term corresponds to the effective bias, and the second term contributes to the effective variance. A similar reformulation can also be applied to the variance error. Further simplifications are possible due to the fact that $\mathbf{C}_0 = \mathbf{0}$, so the variance term corresponding to the first term in (7.3) is 0. Afterwards, $\mathbf{B}_t$ and $\mathbf{C}_t$ can be further characterized by the analysis similar to Zou et al. (2021).

## 8 EXPERIMENTS

In this section, we verify our theoretical findings with empirical experiments. We present the experiments in the linear regression setting, and experiments on the single-neuron ReLU network (Wu et al., 2023) in Appendix A. We (i) compare the generalization performance of SGD with different schemes, and (ii) explore the impact of the choice of the averaging parameter $\alpha$ on the excess risk of SGD with EMA. We consider the well specified setting (Assumption 3.5) with $\sigma^2 = 1$. The data feature vectors follow the Gaussian distribution $\mathbf{x}_t \sim \mathcal{N}(\mathbf{0}, \mathbf{H})$ where the eigenvalue spectrum of $\mathbf{H}$ is $\lambda_i = i^{-2}$ with $d = 2000$, which is also the experiment setting in Zou et al. (2021); Wu et al. (2022); Li et al. (2023). The centered model weight vector is initialized as a Gaussian random vector $\mathbf{w}_0 - \mathbf{w}_* \sim \mathcal{N}(\mathbf{0}, \mathbf{I})$. According to Theorem 4.1, the learning rate $\delta$ should satisfy $\delta < 1/(\psi \operatorname{tr}(\mathbf{H})) = 2/\pi^2 \approx 0.203$, so we choose $\delta = 0.2$. We also present the ablation studies on the learning rate in Appendix A.2. The total number of iterations is fixed as $N = 3000$. In all experiments, we record both the bias error and the variance error as defined in (7.1). We run each experiment 100 times and plot the averaged bias and variance errors. We also present experiment results with error bars in Appendix A. The experiments are runnable on a PC within minutes.

**Comparison of different averaging schemes.** In the comparison of EMA with other averaging schemes, the averaging parameter of EMA is $\alpha = 0.995$, and $s \in \{100, 200, 500, 1000\}$ in tail averaging. The comparisons of the bias error and the variance error are shown in Figures 2(a) and 2(b), respectively. Although the bias error of SGD with EMA decays slowly at the beginning, it achieves a fast decay rate similar to that of SGD without averaging. However, the bias error of SGD with EMA is far more stable than without averaging, due to the reduced variance of the data feature. The variance error of SGD with EMA remains at a low level though slightly larger than SGD with iterate averaging or tail averaging (because $(1 - \alpha)(N - s) \gg 1$). We observe that the variance error curve of SGD with EMA ($\alpha = 0.995$) intersects the curves of SGD with iterate averaging and

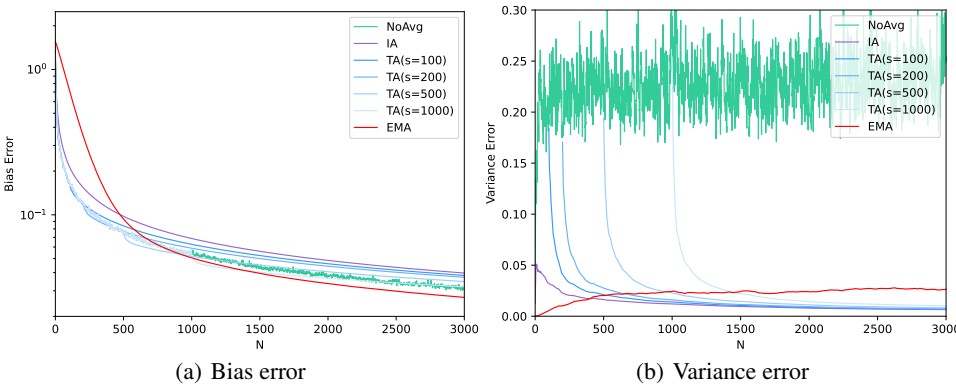

(a) Bias error  (b) Variance error

Figure 2: Comparison of SGD with different averaging schemes. The bias error of SGD with EMA is more stable than SGD without averaging, and decays faster than iterate averaging and tail averaging when $N$ is large. The variance error of SGD with EMA remains relatively small, and is comparable to that of SGD with iterate averaging or tail averaging.

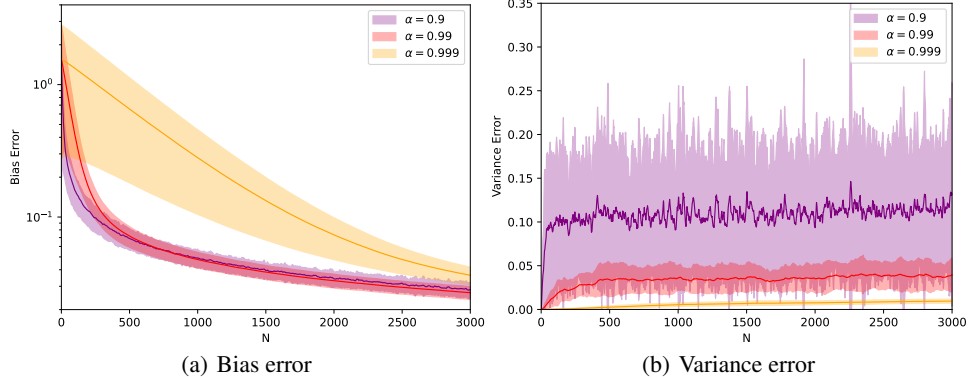

(a) Bias error  (b) Variance error

Figure 3: Comparison of SGD with EMA with different $\alpha$. The bias error of SGD with EMA with smaller alpha decays faster at the beginning of training, but the advantage is less significant when $N$ is large. The variance error of SGD with EMA decreases as $\alpha$ increases.

tail averaging where $s = 100, 200, 500, 1000$ at $N \approx 400, 500, 600, 1000, 15000$, respectively. This verifies the connection between EMA and tail averaging under the condition $(1 - \alpha)(N - s) \simeq 1$. We also conclude that averaging in general is crucial in variance reduction due to the observation that the variance error of SGD with tail averaging decays sharply when averaging starts.

**Comparison of SGD with EMA with different** $\alpha$**.** We compare SGD with EMA with $\alpha = 0.9$, 0.99 and 0.999. The variance error (Figure 3(a)) of SGD with EMA with larger $\alpha$ is significantly smaller than that with smaller $\alpha$, and the bias error (Figure 3(b)) is also more stable. The bias error of SGD with EMA when $\alpha = 0.9$ or 0.99 decays much faster than when $\alpha = 0.999$, but they all approach a similar level when $N = 3000$. We conjecture that this is because the decay rate of the bias error is dominated by the slowest decaying component, which is the bias error in the eigen-subspaces of the smallest eigenvalues. As we have pointed out in Proposition 4.3, the exponential decay rate of the bias error in such eigen-subspaces is independent of $\alpha$.

## 9 CONCLUSION

In this work, we study the generalization of SGD with EMA in the high-dimensional linear regression setting. Our excess risk bound of SGD with EMA depends solely on the eigenvalue spectrum, which is instance-dependent and dimension-free. Similar results can also be derived for mini-batch SGD. In a comparison with SGD with other averaging schemes, we reveal the two-fold advantage of SGD with EMA: the exponentially decaying effective bias error and the modest effective variance error. Our analysis provides the framework for the study of a class of averaging schemes we proposed.

## REPRODUCIBILITY STATEMENT

In the numerical experiments, we have described the details of the model, generation of synthetic data, and evaluation metrics in Section 8. The proof of theoretical results are given in Appendix C, and the proof of supporting lemmas are given in subsequent sections.

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

## A   ADDITIONAL EXPERIMENTS

We first show the results corresponding to Figure 2 with confidence bands[1].

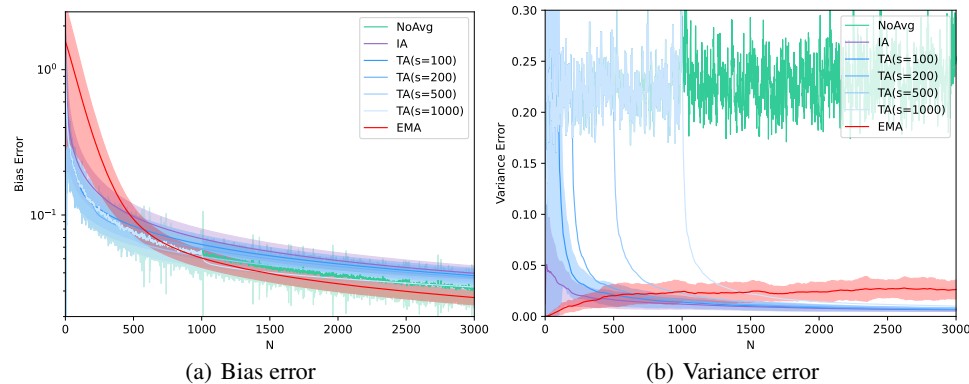

|          (a) Bias error          |          (b) Variance error          |

Figure 4: Comparison of SGD with different averaging schemes, with confidence bands.

### A.1   EXPERIMENTS ON SINGLE-NEURON RELU NETWORKS

In this section, we present additional experiments on the single-neuron ReLU network. The results are similar to the linear regression model.

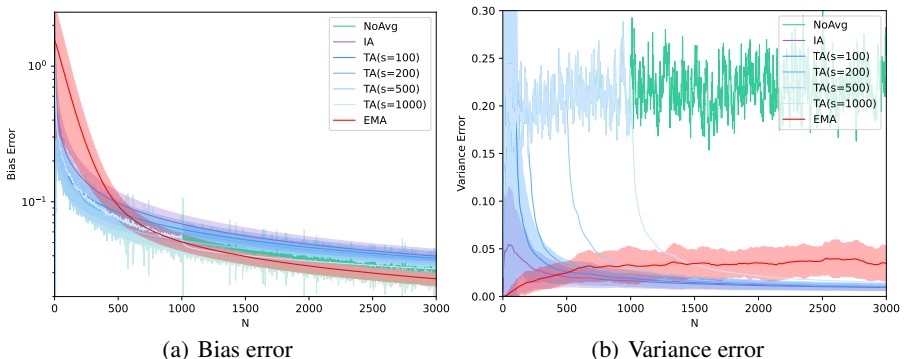

|          (a) Bias error          |          (b) Variance error          |

Figure 5: Comparison of SGD with different averaging schemes for the single-neuron ReLU network.

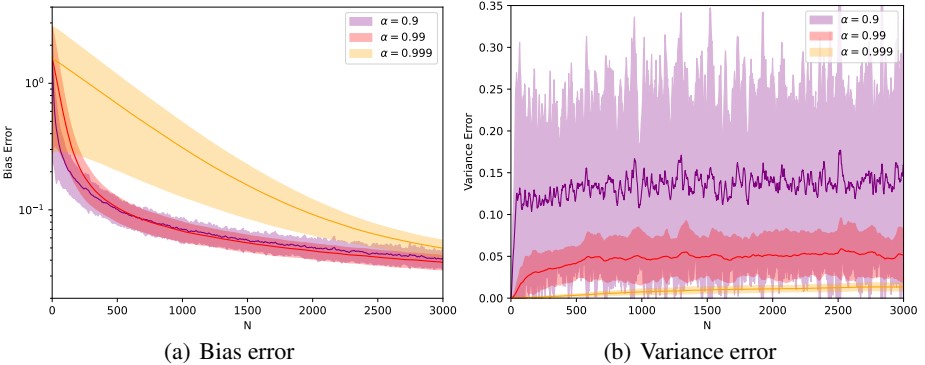

|          (a) Bias error          |          (b) Variance error          |

Figure 6: Comparison of SGD with EMA with different $\alpha$ for the single-neuron ReLU network.

### A.2   ABLATION STUDY ON LEARNING RATE

We next show the effect of the learning rate $\delta$ on the bias and variance errors in the settings of SGD with EMA ($\alpha = 0.995$, Figure 7), iterate averaging (Figure 8), or tail averaging ($s = 500$, Figure 9). We first notice a threshold between $\delta = 0.6$ and $\delta = 0.7$ shared by all averaging schemes,

---

[1]For the variance error, we only show the confidence band for iterate averaging, tail averaging with $s = 100$, and EMA for variance error, for clarity.

beyond which the variance error blows up. It is noticeable this is not the threshold at $\delta = \lambda_1^{-1} = 1$ that causes the **effective variance** to blow up. This justifies the necessity of the condition $\psi \leq 1/(\psi \operatorname{tr}(\mathbf{H}))$ persistent in the excess risk upper bound of all averaging schemes. For all averaging schemes (including EMA), as $\delta$ increases (below the threshold), the bias error decreases, while the variance error increases.

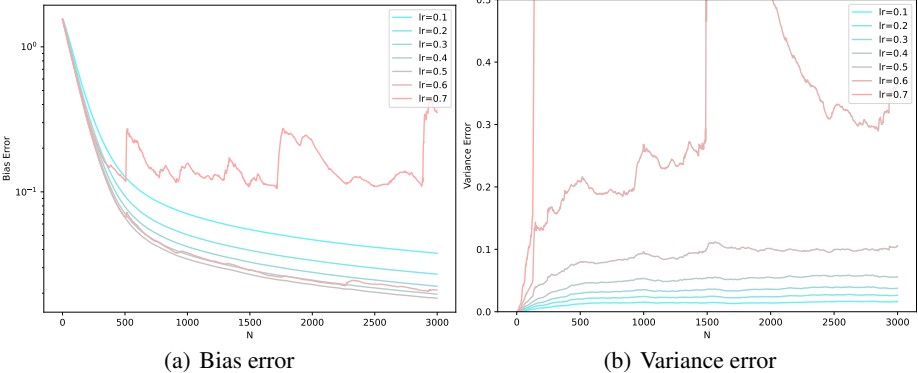

(a) Bias error  (b) Variance error

Figure 7: Comparison of SGD with EMA with different $\delta$.

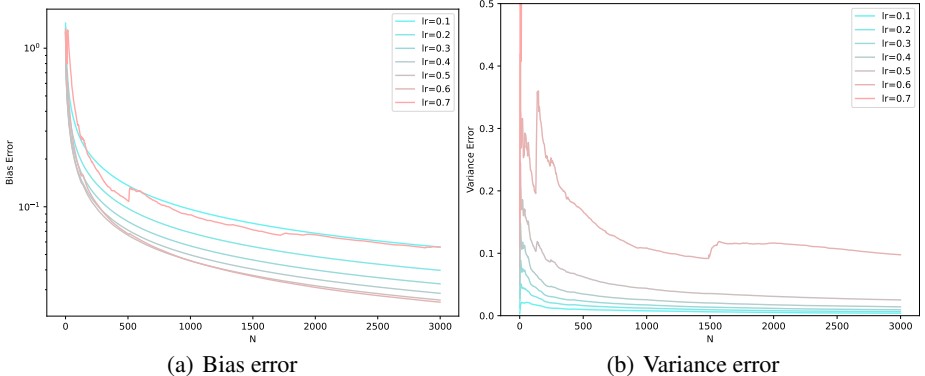

(a) Bias error  (b) Variance error

Figure 8: Comparison of SGD with iterate averaging with different $\delta$.

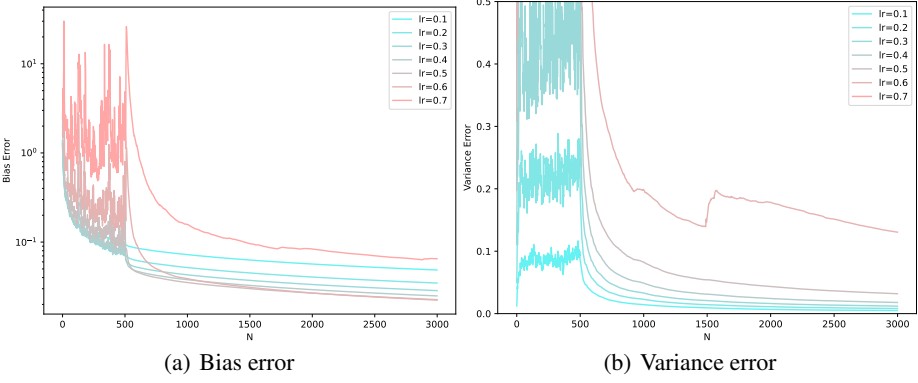

(a) Bias error  (b) Variance error

Figure 9: Comparison of SGD with tail averaging ($s = 500$) with different $\delta$.

## B  ADDITIONAL NOTATIONS

**Linear Operators on Matrices.** We define the following linear operators on matrix following Zou et al. (2021):

$$\mathcal{I} = \mathbf{I} \otimes \mathbf{I}, \qquad \mathcal{M} = \mathbb{E}[\mathbf{x} \otimes \mathbf{x} \otimes \mathbf{x} \otimes \mathbf{x}], \qquad \widetilde{\mathcal{M}} = \mathbf{H} \otimes \mathbf{H},$$

$$\mathcal{B} = \mathbb{E}[(\mathbf{I} - \delta \mathbf{x}\mathbf{x}^\top) \otimes (\mathbf{I} - \delta \mathbf{x}\mathbf{x}^\top)], \qquad \widetilde{\mathcal{B}} = (\mathbf{I} - \delta \mathbf{H}) \otimes (\mathbf{I} - \delta \mathbf{H})$$

Denote the $\sigma$-algebra generated by samples $\{(\mathbf{x}_k, y_k)\}_{k=1}^{t}$ as $\mathcal{F}_t$. Due to the optimality of $\mathbf{w}_*$, we have $\nabla L(\mathbf{w}_*) = \mathbf{0}$, which implies that

$$\mathbf{0} = \nabla L(\mathbf{w}_*) = \mathbb{E}[\mathbf{x}(\mathbf{x}^\top \mathbf{w}_* - y)] = \mathbf{H}\mathbf{w}_* - \mathbb{E}[\mathbf{x} \cdot y]. \tag{B.1}$$

Due to the equality above, we have

$$\mathbb{E}[\boldsymbol{\eta}_t^{\text{bias}}|\mathcal{F}_{t-1}] = (\mathbf{I} - \delta\mathbf{H})\boldsymbol{\eta}_{t-1}^{\text{bias}}, \qquad \mathbb{E}[\boldsymbol{\eta}_t^{\text{var}}|\mathcal{F}_{t-1}] = (\mathbf{I} - \delta\mathbf{H})\boldsymbol{\eta}_{t-1}^{\text{var}}.$$

Iterating this property, using the double expectation formula, we have for any $k \leq t$, we have

$$\mathbb{E}[\boldsymbol{\eta}_t^{\text{bias}}|\mathcal{F}_k] = (\mathbf{I} - \delta\mathbf{H})^{t-k}\boldsymbol{\eta}_k^{\text{bias}}, \qquad \mathbb{E}[\boldsymbol{\eta}_t^{\text{var}}|\mathcal{F}_k] = (\mathbf{I} - \delta\mathbf{H})^{t-k}\boldsymbol{\eta}_k^{\text{var}}, \tag{B.2}$$

which indicates that $\mathbb{E}[\boldsymbol{\eta}_t^{\text{var}}] = \mathbf{0}$. We also have

$$\begin{aligned}
\mathbf{B}_t &= \mathbb{E}[\mathbb{E}[\boldsymbol{\eta}_t^{\text{bias}} \otimes \boldsymbol{\eta}_t^{\text{bias}}|\mathcal{F}_{t-1}]] \\
&= \mathbb{E}[\mathbb{E}[((\mathbf{I} - \delta\mathbf{x}_t\mathbf{x}_t^\top) \otimes (\mathbf{I} - \delta\mathbf{x}_t\mathbf{x}_t^\top)) \cdot (\boldsymbol{\eta}_{t-1}^{\text{bias}} \otimes \boldsymbol{\eta}_{t-1}^{\text{bias}})|\mathcal{F}_{t-1}]] \\
&= \mathbb{E}[\mathcal{B} \circ (\boldsymbol{\eta}_{t-1}^{\text{bias}} \otimes \boldsymbol{\eta}_{t-1}^{\text{bias}})] \\
&= \mathcal{B} \circ \mathbf{B}_{t-1}, 
\end{aligned} \tag{B.3}$$

and

$$\begin{aligned}
\mathbf{C}_t &= \mathbb{E}[\mathbb{E}[\boldsymbol{\eta}_t^{\text{var}} \otimes \boldsymbol{\eta}_t^{\text{var}}|\mathcal{F}_{t-1}]] \\
&= \mathbb{E}\Big[\mathbb{E}[((\mathbf{I} - \delta\mathbf{x}_t\mathbf{x}_t^\top) \otimes (\mathbf{I} - \delta\mathbf{x}_t\mathbf{x}_t^\top)) \cdot (\boldsymbol{\eta}_{t-1}^{\text{var}} \otimes \boldsymbol{\eta}_{t-1}^{\text{var}}) + \delta^2\xi_t^2\mathbf{x}_t\mathbf{x}_t^\top \\
&\qquad - \delta\xi_t\mathbf{x}_t(\boldsymbol{\eta}_{t-1}^{\text{var}})^\top(\mathbf{I} - \delta\mathbf{x}_t\mathbf{x}_t^\top) - \delta\xi_t(\mathbf{I} - \delta\mathbf{x}_t\mathbf{x}_t^\top)\boldsymbol{\eta}_{t-1}^{\text{var}}\mathbf{x}_t^\top|\mathcal{F}_{t-1}]\Big] \\
&= \mathcal{B} \circ \mathbf{C}_{t-1} + \delta^2\boldsymbol{\Sigma}, 
\end{aligned} \tag{B.4}$$

where the last equality holds because $(\mathbf{x}_t, y_t)$ is independent from $\boldsymbol{\eta}_{t-1}^{\text{var}}$ and $\mathbb{E}[\boldsymbol{\eta}_{t-1}^{\text{var}}] = \mathbf{0}$.

Several other key properties of the centered iterates and the linear operators are given in Appendix G.

# C  PROOF OF MAIN RESULTS

## C.1  PROOF OF THEOREM 4.1

To prove Theorem 4.1, we first decompose the excess risk into the bias error and the variance error (Lemma C.1), and then bound them separately (Lemma C.2 and Lemma C.3).

**Lemma C.1.** The excess risk can be decomposed as

$$\mathbb{E}[L(\overline{\mathbf{w}}_N)] - L(\mathbf{w}_*) \leq \text{bias} + \text{var},$$

where

$$\text{bias} = \langle\mathbf{H}, \mathbb{E}[\overline{\boldsymbol{\eta}}_N^{\text{bias}} \otimes \overline{\boldsymbol{\eta}}_N^{\text{bias}}]\rangle, \qquad \text{var} = \langle\mathbf{H}, \mathbb{E}[\overline{\boldsymbol{\eta}}_N^{\text{var}} \otimes \overline{\boldsymbol{\eta}}_N^{\text{var}}]\rangle.$$

**Lemma C.2.** Suppose that Assumption 3.2 holds, and the learning rate satisfies $\delta \leq 1/(\psi\,\text{tr}(\mathbf{H}))$. Then the variance error satisfies

$$\text{var} \leq \frac{\sigma^2}{1 - \psi\delta\,\text{tr}(\mathbf{H})}\left[(1 - \alpha)k^* + \delta\sum_{i=k^*}^{k^\dagger}\lambda_i + N\delta^2\sum_{i>k^\dagger}\lambda_i^2\right].$$

**Lemma C.3.** Suppose that Assumption 3.2 holds, and the learning rate satisfies $\delta \leq 1/(\psi\,\text{tr}(\mathbf{H}))$. Then the bias error satisfies

$$\begin{aligned}
\text{bias} &\leq \frac{\psi(\|\mathbf{w}_0 - \mathbf{w}_*\|_{\mathbf{I}_{0:k^\dagger}}^2 + N\delta\|\mathbf{w}_0 - \mathbf{w}_*\|_{\mathbf{H}_{k^\dagger:\infty}}^2)}{\delta(1 - \psi\delta\,\text{tr}(\mathbf{H}))}\left(k^*(1-\alpha)^2 + \delta^2\sum_{i>k^*}\lambda_i^2\right) \\
&\quad + \sum_{i=1}^{n}(\mathbf{w}_0 - \mathbf{w}_*)_i^2\lambda_i\left[\frac{(\delta\lambda_i)\alpha^N - (1-\alpha)(1-\delta\lambda_i)^N}{\delta\lambda_i - (1-\alpha)}\right]^2.
\end{aligned}$$

## C.2 Proof of Theorem 4.2

The lower bound can be proved using the bias-variance decomposition similar to proof of the upper bound.

**Lemma C.4.** Under Assumption 3.5, the excess risk can be decomposed as

$$\mathbb{E}[L(\overline{\mathbf{w}}_N)] - L(\mathbf{w}_*) = \frac{1}{2}(\text{bias} + \text{var}).$$

**Lemma C.5.** Assume that the hyperparameters satisfy $\delta \leq 1/\lambda_i$, $N \geq 2$ and $\alpha^{N-1} \leq 1/N$. Then the variance error satisfies

$$\text{var} \geq \sigma^2 \left[ \frac{3\alpha^2(1-\alpha)k^*}{16} + \frac{\delta}{100} \sum_{i=k^*+1}^{k^\dagger} \lambda_i + \frac{N\delta^2}{180} \sum_{i>k^\dagger} \lambda_i^2 \right].$$

**Lemma C.6.** Under the same assumptions as Lemma C.5, the bias error satisfies

$$\text{bias} \geq \beta e^{-2} \|\boldsymbol{\eta}_0\|_{\mathbf{H}_{k^\dagger:\infty}}^2 \left[ \frac{3\alpha^2(1-\alpha)k^*}{16} + \frac{\delta}{100} \sum_{i=k^*+1}^{k^\dagger} \lambda_i + \frac{N\delta^2}{180} \sum_{i>k^\dagger} \lambda_i^2 \right]$$

$$+ \sum_{i=1}^{d} \eta_{0,i}^2 \lambda_i \left[ \frac{(\delta\lambda_i)\alpha^N - (1-\alpha)(1-\delta\lambda_i)^N}{\delta\lambda_i - (1-\alpha)} \right]^2.$$

The proofs of Lemma C.1 and Lemma C.4 are given in Appendix E.1. The proofs of Lemma C.2 and Lemma C.5 are given in Appendix E.2. The proofs of Lemma C.3 and Lemma C.6 are given in Appendix E.3.

## C.3 Proof of Theorem 6.1

In this subsection, we modify the proof of Theorem 4.1 to derive the excess risk upper bound for mini-batch SGD.

*Proof of Theorem 6.1.* Define the residual vector of mini-batch SGD in the same way as SGD. We then define the bias and variance residual vectors as

$$\boldsymbol{\eta}_0^{\text{bias}} = \boldsymbol{\eta}_0, \qquad \boldsymbol{\eta}_t^{\text{bias}} = \left( \mathbf{I} - \frac{\delta}{B} \sum_{i=1}^{B} \mathbf{x}_{t,i} \mathbf{x}_{t,i}^\top \right) \boldsymbol{\eta}_{t-1}^{\text{bias}};$$

$$\boldsymbol{\eta}_0^{\text{var}} = \mathbf{0}, \qquad \boldsymbol{\eta}_t^{\text{var}} = \left( \mathbf{I} - \frac{\delta}{B} \sum_{i=1}^{B} \mathbf{x}_{t,i} \mathbf{x}_{t,i}^\top \right) \boldsymbol{\eta}_{t-1}^{\text{var}} + \frac{\delta}{B} \sum_{i=1}^{B} \xi_{t,i} \mathbf{x}_{t,i}.$$

We define the exponential moving average of the bias and variance residual vectors as well as the second moment matrices $\mathbf{B}_t$ and $\mathbf{C}_t$ in the same way as SGD. We then have the bias-variance decomposition lemma similar to Lemma C.1.

We define all linear matrix operators in the same way as SGD except for $\mathcal{B}$, which is defined as

$$\mathcal{B} := \mathbb{E}\left[ \left( \mathbf{I} - \frac{\delta}{B} \sum_{i=1}^{B} \mathbf{x}_{t,i} \mathbf{x}_{t,i}^\top \right) \otimes \left( \mathbf{I} - \frac{\delta}{B} \sum_{i=1}^{B} \mathbf{x}_{t,i} \mathbf{x}_{t,i}^\top \right) \right],$$

then $\mathbf{B}_t$ and $\mathbf{C}_t$ satisfy the following recursive formulas:

$$\mathbf{B}_{t+1} = \mathcal{B} \circ \mathbf{B}_t, \qquad \mathbf{C}_{t+1} = \mathcal{B} \circ \mathbf{C}_t + \frac{\delta^2}{B} \boldsymbol{\Sigma}.$$

We also note that $\mathcal{B} - \widetilde{\mathcal{B}} = \delta^2/B \cdot (\mathcal{M} - \widetilde{\mathcal{M}})$ is still a PSD operator, and for any PSD matrix $\mathbf{A}$, we have

$$(\mathcal{B} - \widetilde{\mathcal{B}}) \circ \mathbf{A} \preceq \frac{\psi\delta^2}{B} \operatorname{tr}(\mathbf{H}\mathbf{A})\mathbf{H}.$$

Therefore, we can substitute the parameters in Theorem 4.1 as $\sigma^2 \leftarrow \sigma^2/B$ and $\psi \leftarrow \psi/B$, and obtain the upper bound for the excess risk of mini-batch SGD. $\qquad\square$

## D  DISCUSSION ABOUT DECAY RATE OF BIAS ERROR

In this section, we study the term

$$
b_i = \alpha^N + (1 - \alpha) \sum_{k=0}^{N-1} \alpha^{N-1-k} (1 - \delta\lambda_i)^k
$$

$$
= \frac{(\delta\lambda_i)\alpha^N - (1 - \alpha)(1 - \delta\lambda_i)^N}{\delta\lambda_i - (1 - \alpha)}
$$

$$
= (1 - \delta\lambda_i)^N + (\delta\lambda_i) \sum_{k=0}^{N-1} \alpha^{N-1-k}(1 - \delta\lambda_i)^k.
$$

To upper bound $b_i$, when $i \leq k^*$, i.e., $1 - \delta\lambda_i \leq \alpha$, we have

$$
b_i = \frac{(\delta\lambda_i)\alpha^N - (1 - \alpha)(1 - \delta\lambda_i)^N}{\delta\lambda_i - (1 - \alpha)} \leq \frac{\delta\lambda_i}{\delta\lambda_i - (1 - \alpha)} \alpha^N,
$$

where the inequality holds because $(1 - \alpha)(1 - \delta\lambda_i)^N \geq 0$. We also have

$$
b_i = \alpha^N + (1 - \alpha) \sum_{k=0}^{N-1} \alpha^{N-1-k}(1 - \delta\lambda_i)^k
$$

$$
\leq \alpha^N + (1 - \alpha) \sum_{k=0}^{N} \alpha^{N-1-k}\alpha^k = \alpha^N + N(1 - \alpha)\alpha^{N-1},
$$

where the inequality holds because $1 - \delta\lambda_i \leq \alpha$.

When $i > k^*$, i.e., $1 - \delta\lambda_i > \alpha$, we have

$$
b_i = (1 - \delta\lambda_i)^N + (\delta\lambda_i) \sum_{k=0}^{N-1} \alpha^{N-1-k} \cdot (1 - \delta\lambda_i)^k
$$

$$
\leq (1 - \delta\lambda_i)^N + (\delta\lambda_i) \sum_{k=0}^{N-1} (1 - \delta\lambda_i)^{N-1-k} \cdot (1 - \delta\lambda_i)^k
$$

$$
= (1 - \delta\lambda_i)^N + N\delta\lambda_i(1 - \delta\lambda_i)^{N-1},
$$

where the inequality holds because $\alpha \leq 1 - \delta\lambda_i$. We also have

$$
b_i = \frac{(1 - \alpha)(1 - \delta\lambda_i)^N - (\delta\lambda_i)\alpha^N}{1 - \alpha - \delta\lambda_i} \leq \frac{1 - \alpha}{1 - \alpha - \delta\lambda_i}(1 - \delta\lambda_i)^N,
$$

where the inequality holds because $(\delta\lambda_i)\alpha^N \geq 0$.

To lower bound $b_i$, we consider the following cases:

**Case 1.** When $(1 - \delta\lambda_i)/\alpha \leq 1 - 1/N$, we have

$$
b_i = \frac{(\delta\lambda_i)\alpha^N - (1 - \alpha)(1 - \delta\lambda_i)^N}{\delta\lambda_i - (1 - \alpha)} \geq \frac{\delta\lambda_i(\alpha^N - (1 - \delta\lambda_i)^N)}{\delta\lambda_i - (1 - \alpha)}
$$

$$
\geq \frac{(\delta\lambda_i)(1 - (1 - 1/N)^N)}{\delta\lambda_i - (1 - \alpha)}\alpha^N \geq \frac{(1 - e^{-1})\delta\lambda_i}{\delta\lambda_i - (1 - \alpha)}\alpha^N,
$$

where the first inequality holds because $1 - \alpha \leq \delta\lambda_i$, the second inequality holds because $1 - \delta\lambda_i/\alpha \leq 1 - 1/N$, and the last inequality holds because $(1 - 1/N)^N \leq 1/e$.

**Case 2.** When $1 - 1/N < (1 - \delta\lambda_i)/\alpha \leq 1$, we have

$$
b_i \geq \alpha^N + (1 - \alpha) \sum_{k=0}^{N-1} \alpha^N(1 - 1/N)^k = \alpha^N + (1 - \alpha)\alpha^{N-1} \cdot N(1 - (1 - 1/N)^N)
$$

$$
\geq \alpha^N + (1 - e^{-1})(1 - \alpha)N\alpha^{N-1},
$$

where the first inequality holds because $1 - \delta\lambda_i \geq (1 - 1/N)\alpha$, and the second inequality holds because $(1 - 1/N)^N \leq 1/e$.

**Case 3.** When $1 < (1 - \delta\lambda_i)/\alpha \leq N/(N-1)$, similar to Case 2, we have

$$b_i \geq (1 - \delta\lambda_i)^N (1 - e^{-1}) N \delta\lambda_i (1 - \delta\lambda_i)^{N-1}.$$

**Case 4.** When $(1 - \delta\lambda_i)/\alpha > N/(N-1)$, similar to Case 1, we have

$$b_i \geq \frac{(1 - e^{-1})(1 - \alpha)}{1 - \alpha - \delta\lambda_i}(1 - \delta\lambda_i)^N.$$

## E  PROOF OF LEMMAS IN APPENDIX C

### E.1  BIAS-VARIANCE DECOMPOSITION

In this subsection, we prove Lemma C.1 and Lemma C.4. The proof is similar to Zou et al. (2021), and is presented here for completeness.

*Proof of Lemma C.1.* By Lemma G.2, the excess risk can be written as

$$\begin{aligned}
&\mathbb{E}[L(\overline{\mathbf{w}}_N)] - L(\mathbf{w}_*)\\
&= \frac{1}{2}\langle \mathbf{H}, \mathbb{E}[\overline{\boldsymbol{\eta}}_N \otimes \overline{\boldsymbol{\eta}}_N]\rangle\\
&= \frac{1}{2}\mathbb{E}\Big[\langle \mathbf{H}, (\overline{\boldsymbol{\eta}}_N^{\text{bias}} + \overline{\boldsymbol{\eta}}_N^{\text{var}}) \otimes (\overline{\boldsymbol{\eta}}_N^{\text{bias}} + \overline{\boldsymbol{\eta}}_N^{\text{var}})\rangle\Big]\\
&\leq \frac{1}{2}\mathbb{E}\Big[\mathbf{H}, (\overline{\boldsymbol{\eta}}_N^{\text{bias}} + \overline{\boldsymbol{\eta}}_N^{\text{var}}) \otimes (\overline{\boldsymbol{\eta}}_N^{\text{bias}} + \overline{\boldsymbol{\eta}}_N^{\text{var}}) + \langle \mathbf{H}, (\overline{\boldsymbol{\eta}}_N^{\text{bias}} - \overline{\boldsymbol{\eta}}_N^{\text{var}}) \otimes (\overline{\boldsymbol{\eta}}_N^{\text{bias}} - \overline{\boldsymbol{\eta}}_N^{\text{var}})\rangle\Big]\\
&= \langle \mathbf{H}, \mathbb{E}[\overline{\boldsymbol{\eta}}_N^{\text{bias}} \otimes \overline{\boldsymbol{\eta}}_N^{\text{bias}}]\rangle + \langle \mathbf{H}, \overline{\boldsymbol{\eta}}_N^{\text{var}} \otimes \overline{\boldsymbol{\eta}}_N^{\text{var}}\rangle\\
&= \text{bias} + \text{var},
\end{aligned}$$

where the second equality holds due to Lemma G.3, and the inequality holds because a positive term is added. □

*Proof of Lemma C.4.* By Lemma G.3, the excess risk can be written as

$$\begin{aligned}
\mathbb{E}[L(\overline{\mathbf{w}}_N)] - L(\mathbf{w}_*) &= \frac{1}{2}\mathbb{E}\Big[\langle \mathbf{H}, (\overline{\boldsymbol{\eta}}_N^{\text{bias}} + \overline{\boldsymbol{\eta}}_N^{\text{var}}) \otimes (\overline{\boldsymbol{\eta}}_N^{\text{bias}} + \overline{\boldsymbol{\eta}}_N^{\text{var}})\rangle\Big]\\
&= \frac{1}{2}\langle \mathbf{H}, \mathbb{E}[\overline{\boldsymbol{\eta}}_N^{\text{bias}} \otimes \overline{\boldsymbol{\eta}}_N^{\text{bias}}]\rangle + \frac{1}{2}\langle \mathbf{H}, \mathbb{E}[\overline{\boldsymbol{\eta}}_N^{\text{var}} \otimes \overline{\boldsymbol{\eta}}_N^{\text{var}}]\rangle + \langle \mathbf{H}, \mathbb{E}[\overline{\boldsymbol{\eta}}_N^{\text{var}} \otimes \overline{\boldsymbol{\eta}}_N^{\text{bias}}]\rangle.
\end{aligned}$$

It then suffices to show that $\mathbb{E}[\overline{\boldsymbol{\eta}}_N^{\text{var}} \otimes \overline{\boldsymbol{\eta}}_N^{\text{bias}}] = \mathbf{0}$, and it further suffices to prove that $\mathbb{E}[\boldsymbol{\eta}_t^{\text{var}} \otimes \boldsymbol{\eta}_s^{\text{bias}}] = \mathbf{0}$ for all $t$ and $s$. According to the recursive formulas of the residual vectors, we have

$$\boldsymbol{\eta}_t^{\text{var}} = \delta \sum_{k=1}^{t} \prod_{l=k+1}^{t} (\mathbf{I} - \delta\mathbf{x}_l\mathbf{x}_l^\top)(\xi_k\mathbf{x}_k),$$

$$\boldsymbol{\eta}_s^{\text{bias}} = \prod_{j=1}^{s}(\mathbf{I} - \delta\mathbf{x}_j\mathbf{x}_j^\top)\boldsymbol{\eta}_0.$$

We then have

$$\mathbb{E}[\boldsymbol{\eta}_t^{\text{var}} \otimes \boldsymbol{\eta}_s^{\text{bias}}] = \delta \sum_{k=1}^{t} \mathbb{E}\Big[\Big(\prod_{l=k+1}^{t}(\mathbf{I} - \delta\mathbf{x}_l\mathbf{x}_l^\top)(\xi_k\mathbf{x}_k)\Big) \otimes \Big(\prod_{j=1}^{s}(\mathbf{I} - \delta\mathbf{x}_j\mathbf{x}_j^\top)\boldsymbol{\eta}_0\Big)\Big] = \mathbf{0},$$

where the second inequality holds because $\xi_k$ is zero-mean and independent of feature vectors (Assumption 3.5). □

### E.2  VARIANCE BOUND

We need the following lemma to prove Lemma C.2.

**Lemma E.1.** Suppose that $\delta \leq 1/(\psi \operatorname{tr}(\mathbf{H}))$. Then for any $t \geq 0$, the inner product of $\mathbf{C}_t$ and $\mathbf{H}$ is upper bounded by

$$\operatorname{tr}(\mathbf{H}\mathbf{C}_t) \leq \frac{\sigma^2 \delta \operatorname{tr}(\mathbf{H})}{1 - \psi \delta \operatorname{tr}(\mathbf{H})}.$$

The proof of Lemma E.1 is given in Appendix F.1. We now provide the proof for Lemma C.2.

*Proof of Lemma C.2.* According to the definition of var and Lemma G.4, we have

$$\operatorname{var} = (1-\alpha)^2 \sum_{t=0}^{N-1} \left\langle \mathbf{H}, \left( \sum_{k=0}^{N-2-t} \alpha^{N-2-t-k}(\mathbf{I}-\delta\mathbf{H})^k \right)((\mathcal{B}-\widetilde{\mathcal{B}}) \circ \mathbf{C}_t + \delta^2 \boldsymbol{\Sigma}) \right.$$

$$\left. \cdot \left( \sum_{k=0}^{N-2-t} \alpha^{N-2-t-k}(\mathbf{I}-\delta\mathbf{H})^k \right) \right\rangle$$

$$\leq \sum_{t=0}^{N-1} (1-\alpha)^2 \delta^2 (\psi \operatorname{tr}(\mathbf{H}\mathbf{C}_t) + \sigma^2)$$

$$\cdot \left\langle \mathbf{H}, \left( \sum_{k=0}^{N-2-t} \alpha^{N-2-t-k}(\mathbf{I}-\delta\mathbf{H})^k \right) \mathbf{H} \left( \sum_{k=0}^{N-2-t} \alpha^{N-2-t-k}(\mathbf{I}-\delta\mathbf{H})^k \right) \right\rangle$$

$$\leq \frac{\sigma^2}{1-\psi\delta \operatorname{tr}(\mathbf{H})} \sum_{i=1}^{d} \underbrace{(1-\alpha)^2 (\delta\lambda_i)^2 \sum_{t=0}^{N-1} \left( \sum_{k=0}^{N-2-t} \alpha^{N-2-t-k}(1-\delta\lambda_i)^k \right)^2}_{J_i}, \qquad (\text{E.1})$$

where the first inequality holds due to Lemma G.1 part b and Assumption 3.4, and the second inequality holds due to Lemma E.1. We then study the upper bound for $J_i$. Firstly, we have

$$J_i \leq (1-\alpha)^2 (\delta\lambda_i)^2 \sum_{t=0}^{\infty} \left( \sum_{k=0}^{N-2-t} \alpha^{N-2-t-k}(1-\delta\lambda_i)^k \right)^2$$

$$= \frac{(1-\alpha)\delta\lambda_i}{1-\alpha+\alpha\delta\lambda_i} \cdot \frac{1+\alpha-\alpha\delta\lambda_i}{(1+\alpha)(2-\delta\lambda_i)}$$

$$\leq \frac{(1-\alpha)\delta\lambda_i}{1-\alpha+\alpha\delta\lambda_i} \cdot 1$$

$$\leq \min\{1-\alpha, \delta\lambda_i\}, \qquad (\text{E.2})$$

where the first inequality holds because positive terms are added, the second inequality holds because $1+\alpha-\delta\lambda_i \leq 1+\alpha \leq (1+\alpha)(2-\delta\lambda_i)$, and the second inequality holds because $1-\alpha+\alpha\delta\lambda_i \geq \max\{1-\alpha, \delta\lambda_i\}$. Secondly, we have

$$J_i \leq (1-\alpha)^2 (\delta\lambda_i)^2 \sum_{t=0}^{N-1} \left( \sum_{k=0}^{N-2-t} \alpha^{N-2-t-k} \right)^2 = (\delta\lambda_i)^2 \sum_{t=0}^{N-1} (1-\alpha^{N-t-1})^2 \leq N\delta^2\lambda_i^2, \quad (\text{E.3})$$

where the first inequality holds because $1-\delta\lambda_i \leq 1$, and the second inequality holds because $1-\alpha^{N-1-t} \leq 1$. Substituting (E.2) and (E.3) into (E.1), we have

$$\operatorname{var} \leq \frac{\sigma^2}{1-\psi\delta \operatorname{tr}(\mathbf{H})} \sum_{i=1}^{d} \min\left\{1-\alpha, \delta\lambda_i, N\delta^2\lambda_i^2\right\}$$

$$= \frac{\sigma^2}{1-\psi\delta \operatorname{tr}(\mathbf{H})} \left[ (1-\alpha)k^* + \delta \sum_{i=k^*+1}^{k^\dagger} \lambda_i + N\delta^2 \sum_{i>k^\dagger} \lambda_i^2 \right].$$

$\square$

*Proof of Lemma C.5.* According to the definition of var and Lemma G.4, we have

$$\operatorname{var} = (1-\alpha)^2 \sum_{t=0}^{N-1} \left\langle \mathbf{H}, \left( \sum_{k=0}^{N-2-t} \alpha^{N-2-t-k}(\mathbf{I}-\delta\mathbf{H})^k \right)((\mathcal{B}-\widetilde{\mathcal{B}}) \circ \mathbf{C}_t + \delta^2 \boldsymbol{\Sigma}) \right.$$

$$\left( \sum_{k=0}^{N-2-t} \alpha^{N-2-t-k}(\mathbf{I} - \delta\mathbf{H})^k \right) \Big\rangle$$

$$\geq \sigma^2 \sum_{t=0}^{N-1} (1-\alpha)^2 \delta^2 \Big\langle \mathbf{H}, \left( \sum_{k=0}^{N-2-t} \alpha^{N-2-t-k}(\mathbf{I} - \delta\mathbf{H})^k \right) \mathbf{H} \left( \sum_{k=0}^{N-2-t} \alpha^{N-2-t-k}(\mathbf{I} - \delta\mathbf{H})^k \right) \Big\rangle$$

$$= \sigma^2 \sum_{i=1}^{d} \underbrace{(1-\alpha)^2 (\delta\lambda_i)^2 \sum_{t=0}^{N-1} \left( \sum_{k=0}^{t-1} \alpha^{t-1-k}(1-\delta\lambda_i)^k \right)^2}_{J_i},$$

where the inequality holds due to Lemma G.1 part b. We then study the lower bound for $J_i$, based on the regime that $\lambda_i$ falls into:

**Case 1:** $i \leq k^*$. In this case, $1 - \delta\lambda_i \leq \alpha$, and we have

$$J_i = \frac{\delta\lambda_i(1-\alpha)(1+\alpha-\alpha\delta\lambda_i)(1-\alpha^{2N})}{(1-\alpha+\alpha\delta\lambda_i)(1+\alpha)(2-\delta\lambda_i)}$$

$$- \frac{2\delta\lambda_i(1-\alpha)^2(1-\delta\lambda_i)\alpha^N}{(1-\alpha+\alpha\delta\lambda_i)(2-\delta\lambda_i)} \cdot \frac{\alpha^N - (1-\delta\lambda_i)^N}{\alpha - (1-\delta\lambda_i)} - \frac{(1-\alpha)^2\delta\lambda_i}{2-\delta\lambda_i}\left( \frac{\alpha^N - (1-\delta\lambda_i)^N}{\alpha - (1-\delta\lambda_i)} \right)^2$$

$$\geq \frac{\delta\lambda_i(1-\alpha)(1+\alpha-\alpha\delta\lambda_i)(1-\alpha^{2N})}{(1-\alpha+\alpha\delta\lambda_i)(1+\alpha)(2-\delta\lambda_i)} - \frac{2\delta\lambda_i(1-\alpha)^2(1-\delta\lambda_i)\alpha^N}{(1-\alpha+\alpha\delta\lambda_i)(2-\delta\lambda_i)} - \frac{\delta\lambda_i(1-\alpha)^2}{2-\delta\lambda_i}$$

$$= \frac{\delta\lambda_i(1-\alpha)(1+\alpha-\alpha\delta\lambda_i)(\alpha^2-\alpha^{2N})}{(1-\alpha+\alpha\delta\lambda_i)(1+\alpha)(2-\delta\lambda_i)} + \frac{2\delta\lambda_i(1-\alpha)^2(1-\delta\lambda_i)(\alpha-\alpha^N)}{(1-\alpha+\alpha\delta\lambda_i)(2-\delta\lambda_i)}$$

$$\geq \frac{\delta\lambda_i(1-\alpha)(1+\alpha-\alpha\delta\lambda_i)(\alpha^2-\alpha^{2N})}{(1-\alpha+\alpha\delta\lambda_i)(1+\alpha)(2-\delta\lambda_i)},$$

where the first inequality holds because $\frac{\alpha^N - (1-\delta\lambda_i)^N}{\alpha - (1-\delta\lambda_i)} \leq N\alpha^{N-1} \leq 1$, and the second inequality holds because a positive term is dropped. We then consider the function

$$f(x) = \frac{(1-x)(1+\alpha x)}{(1-\alpha x)(1+x)} = 1 - \frac{2(1-\alpha)}{1/x - \alpha x + (1-\alpha)}, \qquad x \in (0, \alpha],$$

so $f(x)$ is decreasing in $x$, and $f(x) \geq f(\alpha) = (1+\alpha^2)/(1+\alpha)^2 \geq 1/2$ (Cauchy-Schwarz inequality). Since $\alpha^{N-1} \leq 1/N$, we also have $1 - \alpha^{2(N-1)} \geq 1 - 1/N^2 \geq 3/4$ because $N \geq 2$. We thus have

$$J_i = (1-\alpha) \cdot f(1-\delta\lambda_i) \cdot \frac{\alpha^2(1-\alpha^{2(N-1)})}{1+\alpha}$$

$$\geq (1-\alpha) \cdot \frac{1}{2} \cdot \frac{3\alpha^2}{4(1+\alpha)}$$

$$\geq \frac{3(1-\alpha)\alpha^2}{16},$$

where the last inequality holds because $\alpha \leq 1$.

**Case 2:** $k^* < i \leq k^\dagger$. In this case, $1 - 1/N \leq 1 - \delta\lambda_i \leq \alpha$, and for any $\mu \in (1, N)$, we have

$$J_i \geq (1-\alpha)^2(\delta\lambda_i)^2 \sum_{t=0}^{N-1} \left( (1-\delta\lambda_i)^{t-1} \sum_{k=0}^{t-1} \alpha^k \right)^2$$

$$= (\delta\lambda_i)^2 \sum_{t=0}^{N-1} (1-\delta\lambda_i)^{2(t-1)}(1-\alpha^t)^2$$

$$\geq (\delta\lambda_i)^2 \sum_{t=\lceil \log_{1/\alpha} \mu \rceil}^{N-1} (1-\delta\lambda_i)^{2(t-1)}(1-\alpha^t)^2$$

$$\geq (\delta\lambda_i)^2 (1-1/\mu)^2 \sum_{t=\lceil \log_{1/\alpha} \mu \rceil}^{N-1} (1-\delta\lambda_i)^{2(t-1)}$$

$$= \frac{\delta \lambda_i (1 - 1/\mu)^2}{2 - \delta \lambda_i} [(1 - \delta \lambda_i)^{2(\lceil \log_{1/\alpha} \mu \rceil - 1)} - (1 - \delta \lambda_i)^{2(N-1)}],$$

where the first inequality holds because $(1 - \delta \lambda_i)^k \leq (1 - \delta \lambda_i)^{t-1}$, the second inequality holds because negative terms are dropped, and the last inequality holds because $\alpha^t \leq \alpha^{\lceil \log_{1/\alpha} \mu \rceil} \leq 1/\mu$. Since $1 - \delta \lambda_i \geq \alpha$, we have

$$(1 - \delta \lambda_i)^{2(\lceil \log_{1/\alpha} \mu \rceil - 1)} \geq \alpha^{2(\lceil \log_{1/\alpha} \mu \rceil - 1)} \geq \alpha^{2 \log_{1/\alpha} \mu} = \mu^{-2}.$$

Furthermore, since $1 - \delta \lambda_i \leq 1 - 1/N$, we have

$$(1 - \delta \lambda_i)^{2(N-1)} \leq (1 - 1/N)^{2(N-1)} \leq (1/2)^2 = 1/4,$$

where the second inequality holds because $(1 - 1/N)^{2N-2}$ is decreasing in $N$ when $N \geq 2$. Therefore, by taking $\mu^{-1} = (1 + \sqrt{3})/4$, we have

$$J_i \geq \frac{\delta \lambda_i}{2} \cdot \frac{6\sqrt{3} - 9}{64} \geq \frac{\delta \lambda_i}{100}.$$

**Case 3**: $i > k^\dagger$. In this case $\lambda_i \leq 1/N\delta$, and for all $k < N$, we have

$$(1 - \delta \lambda_i)^k \geq (1 - 1/N)^{N-1} \geq e^{-1},$$

where the second inequality holds because $(1 - 1/N)^{N-1}$ is decreasing in $N$ when $N \geq 2$ and the limit is $e^{-1}$. We then have

$$J_i \geq e^{-2}(1-\alpha)^2(\delta \lambda_i)^2 \sum_{t=0}^{N-1} \left( \sum_{k=0}^{t-1} \alpha^k \right)^2$$

$$= e^{-2}(\delta \lambda_i)^2 \sum_{t=0}^{N-1} (1 - \alpha^t)^2$$

$$\geq e^{-2}(\delta \lambda_i)^2 \sum_{t=\lfloor N/2 \rfloor}^{N-1} (1 - \alpha^t)^2$$

$$\geq \frac{N}{2e^2} \delta^2 \lambda_i^2 (1 - \alpha^{(N-1)/2})^2$$

$$\geq N\delta^2 \lambda_i^2 \cdot \frac{1}{2e^2} \cdot (1 - 1/\sqrt{2})^2 \geq \frac{N\delta^2 \lambda_i^2}{180},$$

where the second inequality holds because positive terms are dropped, the third inequality holds because for all $t \geq \lfloor N/2 \rfloor$, we have $\alpha^t \leq \alpha^{(N-1)/2}$, and the fourth inequality holds because $\alpha^{N-1} \leq 1/N \leq 1/2$.

Combining all the above, we have

$$\text{var} \geq \sigma^2 \left[ \frac{3\alpha^2(1-\alpha)k^*}{16} + \frac{\delta}{100} \sum_{i=k^*+1}^{k^\dagger} \lambda_i + \frac{N\delta^2}{180} \sum_{i>k^\dagger} \lambda_i^2 \right].$$

$\square$

### E.3 BIAS BOUND

We need the following lemma to prove Lemma C.3

**Lemma E.2.** The matrices $\mathbf{B}_t$ satisfies

$$\sum_{k=1}^{t} \text{tr}(\mathbf{H}\mathbf{B}_k) \leq \frac{1}{\delta(1 - \psi \delta \, \text{tr}(\mathbf{H}))} \sum_{i=1}^{d} \eta_{0,i}^2 [1 - (1 - \delta \lambda_i)^t].$$

The proof of Lemma E.2 is given in Appendix F.2. We then prove Lemma C.3.

*Proof of Lemma C.3.* By definition of the bias error and Lemma G.4, we have

$$\text{bias} \leq \psi \sum_{i=1}^{d} \underbrace{(1-\alpha)^2 (\delta\lambda_i)^2 \sum_{t=0}^{N-1} \text{tr}(\mathbf{H}\mathbf{B}_t) \left( \sum_{k=0}^{N-2-t} \alpha^{N-2-t-k} (1-\delta\lambda_i)^k \right)^2}_{J_i}$$

$$+ \sum_{i=1}^{d} \eta_{0,i}^2 \lambda_i \left[ \frac{(1-\alpha)(1-\delta\lambda_i)^N - (\delta\lambda_i)\alpha^N}{1 - \delta\lambda_i - \alpha} \right]^2,$$

where the inequality holds due to Lemma G.1 part b. We then study the upper bound of $J_i$. Firstly, we have

$$J_i \leq (1-\alpha)^2 (\delta\lambda_i)^2 \sum_{t=0}^{N-1} \text{tr}(\mathbf{H}\mathbf{B}_t) \left( \sum_{k=0}^{N-2-t} (1-\delta\lambda_i)^k \right)^2$$

$$= (1-\alpha)^2 \sum_{t=0}^{N-1} \text{tr}(\mathbf{H}\mathbf{B}_t)(1 - (1-\delta\lambda_i)^{N-1-t})^2$$

$$\leq (1-\alpha)^2 \sum_{t=0}^{N-1} \text{tr}(\mathbf{H}\mathbf{B}_t)$$

$$\leq \frac{(1-\alpha)^2}{\delta(1 - \psi\,\text{tr}(\mathbf{H}))} \sum_{i=1}^{d} \eta_{0,i}^2 [1 - (1-\delta\lambda_i)^N]$$

$$\leq \frac{(1-\alpha)^2}{\delta(1 - \psi\,\text{tr}(\mathbf{H}))} (\|\boldsymbol{\eta}_0\|_{\mathbf{I}_{0:k^\dagger}}^2 + N\delta\|\boldsymbol{\eta}_0\|_{\mathbf{H}_{k^\dagger:\infty}}^2),$$

where the first inequality holds because $\alpha \leq 1$, the second inequality holds because $1 - (1 - \delta\lambda_i)^{N-1-t} \leq 1$, the third inequality holds due to Lemma E.2, and the last inequality holds because $1 - (1-\delta\lambda_i)^N \leq \min\{1, N\delta\lambda_i\}$. Secondly, we have

$$J_i \leq (1-\alpha)^2 (\delta\lambda_i)^2 \sum_{t=0}^{N-1} \text{tr}(\mathbf{H}\mathbf{B}_t) \left( \sum_{k=0}^{N-2-t} \alpha^{N-2-t-k} \right)^2$$

$$= (\delta\lambda_i)^2 \sum_{t=0}^{N-1} \text{tr}(\mathbf{H}\mathbf{B}_t)(1 - \alpha^{N-1-t})^2$$

$$\leq (\delta\lambda_i)^2 \sum_{t=0}^{N-1} \text{tr}(\mathbf{H}\mathbf{B}_t)$$

$$\leq \frac{\delta\lambda_i^2}{1 - \psi\delta\,\text{tr}(\mathbf{H})} \sum_{i=1}^{d} \eta_{0,i}^2 [1 - (1-\delta\lambda_i)^N]$$

$$\leq \frac{\delta\lambda_i^2}{1 - \psi\delta\,\text{tr}(\mathbf{H})} (\|\boldsymbol{\eta}_0\|_{\mathbf{I}_{0:k^\dagger}}^2 + N\delta\|\boldsymbol{\eta}_0\|_{\mathbf{H}_{k^\dagger:\infty}}^2),$$

where the first inequality holds because $1 - \delta\lambda_i \leq 1$, the second inequality holds because $1 - \alpha^{N-1-t} \leq 1$, the third inequality holds due to Lemma E.2, and the last inequality holds because $1 - (1-\delta\lambda_i)^N \leq \min\{1, N\delta\lambda_i\}$. Combining all the above, we have

$$\text{bias} \leq \frac{\psi(\|\mathbf{w}_0 - \mathbf{w}_*\|_{\mathbf{I}_{0:k^\dagger}}^2 + N\delta\|\mathbf{w}_0 - \mathbf{w}_*\|_{\mathbf{H}_{k^\dagger:\infty}}^2)}{\delta(1 - \psi\delta\,\text{tr}(\mathbf{H}))} \left( k^*(1-\alpha)^2 + \delta^2 \sum_{i>k^*} \lambda_i^2 \right)$$

$$+ \sum_{i=1}^{n} (\mathbf{w}_0 - \mathbf{w}_*)_i^2 \lambda_i \left[ \frac{(\delta\lambda_i)\alpha^N - (1-\alpha)(1-\delta\lambda_i)^N}{\delta\lambda_i - (1-\alpha)} \right]^2.$$

$\square$

*Proof of Lemma C.6.* According to the definition of the bias error and Lemma G.4, we have

$$\text{bias} \geq \beta \sum_{i=1}^{d} (1-\alpha)^2 (\delta\lambda_i)^2 \sum_{t=0}^{N-1} \text{tr}(\mathbf{H}\mathbf{B}_t) \left( \sum_{k=0}^{N-2-t} \alpha^{N-2-t-k}(1-\delta\lambda_i)^k \right)^2$$

$$+ \sum_{i=1}^{d} \eta_{0,i}^2 \lambda_i \left[ \frac{(\delta\lambda_i)\alpha^N - (1-\alpha)(1-\delta\lambda_i)^N}{\delta\lambda_i - (1-\alpha)} \right]^2$$

$$\geq \beta \, \text{tr}(\mathbf{B}_0 \mathbf{H}(\mathbf{I} - \delta\mathbf{H})^{2(N-1)}) \sum_{i=1}^{d} (1-\alpha)^2 (\delta\lambda_i)^2 \underbrace{\sum_{t=0}^{N-1} \left( \sum_{k=0}^{t-1} \alpha^{t-1-k}(1-\delta\lambda_i)^k \right)^2}_{J_i}$$

$$+ \sum_{i=1}^{d} \eta_{0,i}^2 \lambda_i \left[ \frac{(\delta\lambda_i)\alpha^N - (1-\alpha)(1-\delta\lambda_i)^N}{\delta\lambda_i - (1-\alpha)} \right]^2,$$

where the second inequality holds because

$$\mathbf{B}_t = \mathcal{B}^t \circ \mathbf{B}_0 \succeq \widetilde{\mathcal{B}}^t \circ \mathbf{B}_0 = (\mathbf{I} - \delta\mathbf{H})^t \mathbf{B}_0 (\mathbf{I} - \delta\mathbf{H})^t \succeq (\mathbf{I} - \delta\mathbf{H})^{N-1} \mathbf{B}_0 (\mathbf{I} - \delta\mathbf{H})^{N-1}.$$

Note that the lower bound for $J_i$ is the same as that in the proof of Lemma C.5. For the term $\text{tr}(\mathbf{B}_0 \mathbf{H}(\mathbf{I} - \delta\mathbf{H})^{2N})$, we have

$$\text{tr}(\mathbf{B}_0 \mathbf{H}(\mathbf{I} - \delta\mathbf{H})^{2N}) = \sum_{i=1}^{d} \eta_{0,i}^2 \lambda_i (1-\delta\lambda_i)^{2(N-1)} \geq \sum_{i>k^\dagger} \eta_{0,i}^2 \lambda_i (1-1/N)^{2(N-1)} \geq e^{-2} \|\boldsymbol{\eta}_0\|^2_{\mathbf{H}_{k^\dagger:\infty}},$$

where the second inequality holds because $\delta\lambda_i \leq 1/N$ when $i > k^\dagger$, and the second inequality holds because $(1-1/N)^{2(N-1)} \geq 1/e^2$. We thus have

$$\text{bias} \geq \beta e^{-2} \|\boldsymbol{\eta}_0\|^2_{\mathbf{H}_{k^\dagger:\infty}} \left[ \frac{3\alpha^2(1-\alpha)k^*}{16} + \frac{\delta}{100} \sum_{i=k^*+1}^{k^\dagger} \lambda_i + \frac{N\delta^2}{180} \sum_{i>k^\dagger} \lambda_i^2 \right]$$

$$+ \sum_{i=1}^{d} \eta_{0,i}^2 \lambda_i \left[ \frac{(\delta\lambda_i)\alpha^N - (1-\alpha)(1-\delta\lambda_i)^N}{\delta\lambda_i - (1-\alpha)} \right]^2.$$

$\square$

# F  PROOF OF LEMMAS IN APPENDIX E

## F.1  PROOF OF LEMMA E.1

We need the following lemmas to prove Lemma E.1:

**Lemma F.1.** $\mathbf{C}_t$ satisfies

$$\mathbf{C}_t = \sum_{k=0}^{k-1} \mathcal{B}^k \circ \boldsymbol{\Sigma}.$$

Since $\mathcal{B}$ is a PSD operator (Lemma G.1), we have

$$\mathbf{C}_0 \preceq \mathbf{C}_1 \preceq \cdots \mathbf{C}_t \preceq \cdots.$$

*Proof.* The expression for $\mathbf{C}_t$ follows directly from the recursive formula for $\mathbf{C}_t$. $\square$

We now provide the proof of Lemma E.1.

*Proof of Lemma E.1.* According to the recursive formula, we have

$$\mathbf{C}_t = \mathcal{B} \circ \mathbf{C}_{t-1} + \delta^2 \boldsymbol{\Sigma} \preceq \widetilde{\mathcal{B}} \circ \mathbf{C}_{t-1} + \delta^2 (\psi \, \text{tr}(\mathbf{H}\mathbf{C}_{t-1}) + \sigma^2)\mathbf{H}$$

$$\preceq \sum_{k=0}^{t-1} (\psi\delta^2 \, \text{tr}(\mathbf{H}\mathbf{C}_{t-1-k}) + \sigma^2) \cdot \widetilde{\mathcal{B}}^k \circ \mathbf{H}$$

$$\preceq \delta^2(\psi \operatorname{tr}(\mathbf{HC}_t) + \sigma^2) \sum_{k=0}^{t-1} \widetilde{\mathcal{B}}^k \circ \mathbf{H}$$

$$\preceq \delta^2(\psi \operatorname{tr}(\mathbf{HC}_t) + \sigma^2) \sum_{k=0}^{\infty} \widetilde{\mathcal{B}}^k \circ \mathbf{H},$$

where the first inequality holds due to Lemma G.1 part b and Assumption 3.2, the second inequality holds by recursively applying the first inequality, the third inequality holds due to Lemma F.1, and the last inequality holds because $\widetilde{\mathcal{B}}$ is a PSD operator (Lemma G.1, part a). Taking the inner product with $\mathbf{H}$ on both sides of the inequality, we have

$$\operatorname{tr}(\mathbf{HC}_t) \leq \delta^2(\psi \operatorname{tr}(\mathbf{HC}_t) + \sigma^2) \sum_{k=0}^{\infty} \operatorname{tr}(\mathbf{H}(\mathbf{I} - \delta\mathbf{H})^k \mathbf{H}(\mathbf{I} - \delta\mathbf{H})^k)$$

$$= \delta^2(\psi \operatorname{tr}(\mathbf{HC}_t) + \sigma^2) \sum_{k=0}^{\infty} \operatorname{tr}(\mathbf{H}(\mathbf{I} - \delta\mathbf{H})^{2k} \mathbf{H})$$

$$\leq \delta^2(\psi \operatorname{tr}(\mathbf{HC}_t) + \sigma^2) \sum_{k=0}^{\infty} \operatorname{tr}(\mathbf{H}(\mathbf{I} - \delta\mathbf{H})^k \mathbf{H})$$

$$= \delta(\psi \operatorname{tr}(\mathbf{HC}_t) + \sigma^2) \operatorname{tr}(\mathbf{H}),$$

where the second inequality holds because $\mathbf{I} - \delta\mathbf{H} \succ 0$. Rearranging terms, as long as $\delta < 1/(\psi \operatorname{tr}(\mathbf{H}))$, we have

$$\operatorname{tr}(\mathbf{HC}_t) \leq \frac{\sigma^2 \delta \operatorname{tr}(\mathbf{H})}{1 - \psi\delta \operatorname{tr}(\mathbf{H})}.$$

$\square$

### F.2 PROOF OF LEMMA E.2

*Proof of Lemma E.2.* Define

$$\mathbf{S}_t^1 = \sum_{k=0}^{t-1} \mathbf{B}_t.$$

Note that $\mathbf{S}_t^1$ satisfies $\mathbf{S}_t^1 = \mathcal{B} \circ \mathbf{S}_{t-1} + \mathbf{B}_0$, so according to Lemma G.1 part b, $\mathbf{S}_t^1$ can be bounded by

$$\mathbf{S}_t^1 \preceq \widetilde{\mathcal{B}} \circ \mathbf{S}_{t-1}^1 + \psi\delta^2 \operatorname{tr}(\mathbf{HS}_{t-1}^1)\mathbf{H} + \mathbf{B}_0$$

$$\preceq \sum_{k=0}^{t-1} \widetilde{\mathcal{B}}^k \circ (\psi\delta^2 \operatorname{tr}(\mathbf{HS}_{t-1-k}^1)\mathbf{H} + \mathbf{B}_0)$$

$$\preceq \sum_{k=0}^{t-1} \widetilde{\mathcal{B}}^k \circ (\psi\delta^2 \operatorname{tr}(\mathbf{HS}_t^1)\mathbf{H} + \mathbf{B}_0)$$

$$= \psi\delta^2 \operatorname{tr}(\mathbf{HS}_t^1) \sum_{k=0}^{t-1} (\mathbf{I} - \delta\mathbf{H})^k \mathbf{H}(\mathbf{I} - \delta\mathbf{H})^k + \sum_{k=0}^{t-1} (\mathbf{I} - \delta\mathbf{H})^k \mathbf{B}_0 (\mathbf{I} - \delta\mathbf{H})^k,$$

where the second inequality holds by recursively applying the first inequality, and the third inequality holds because $\mathbf{S}_{t-1-k}^1 \preceq \mathbf{S}_t^1$. Taking the inner produce on both sides of the inequality, we have

$$\operatorname{tr}(\mathbf{HS}_t^1) \leq \psi\delta^2 \operatorname{tr}(\mathbf{HS}_t^1) \sum_{k=0}^{t-1} \operatorname{tr}(\mathbf{H}^2(\mathbf{I} - \delta\mathbf{H})^{2k}) + \sum_{k=0}^{t-1} \operatorname{tr}(\mathbf{B}_0\mathbf{H}(\mathbf{I} - \delta\mathbf{H})^{2k})$$

$$\leq \psi\delta^2 \operatorname{tr}(\mathbf{HS}_t^1) \sum_{k=0}^{t-1} \operatorname{tr}(\mathbf{H}^2(\mathbf{I} - \delta\mathbf{H})^k) + \sum_{k=0}^{t-1} \operatorname{tr}(\mathbf{B}_0\mathbf{H}(\mathbf{I} - \delta\mathbf{H})^k)$$

$$\leq \psi \delta^2 \operatorname{tr}(\mathbf{H} \mathbf{S}_t^1) \sum_{k=0}^{\infty} \operatorname{tr}(\mathbf{H}^2 (\mathbf{I} - \delta \mathbf{H})^k) + \sum_{k=0}^{t-1} \operatorname{tr}(\mathbf{B}_0 \mathbf{H} (\mathbf{I} - \delta \mathbf{H})^k)$$

$$= \psi \delta \operatorname{tr}(\mathbf{H}) \operatorname{tr}(\mathbf{H} \mathbf{S}_t^1) + \delta^{-1} \sum_{i=1}^{d} \eta_{0,i}^2 (1 - (1 - \delta \lambda_i)^t),$$

where the second inequality holds because $(\mathbf{I} - \delta \mathbf{H})^{2k} \preceq (\mathbf{I} - \delta \mathbf{H})^k$, and the third inequality holds because positive terms $\operatorname{tr}(\mathbf{H}^2 (\mathbf{I} - \delta \mathbf{H})^k)$ for $k \geq t$ are added. Rearranging terms, we have

$$\operatorname{tr}(\mathbf{H} \mathbf{S}_t^1) \leq \frac{1}{\delta(1 - \psi \delta \operatorname{tr}(\mathbf{H}))} \sum_{i=1}^{d} \eta_{0,i}^2 [1 - (1 - \delta \lambda_i)^t].$$

$\square$

# G  PROPERTIES OF CENTERED ITERATES AND LINEAR OPERATORS ON MATRICES

**Lemma G.1.** The linear operators on matrix enjoy the following properties:

a. $\mathcal{M}, \widetilde{\mathcal{M}}, \mathcal{B}$, and $\widetilde{\mathcal{B}}$ are all PSD operators, i.e., for any PSD matrix $\mathbf{A}$, we have that $\mathcal{M} \circ \mathbf{A}, \widetilde{\mathcal{M}} \circ \mathbf{A}$, $\mathcal{B} \circ \mathbf{A}$, and $\widetilde{\mathcal{B}} \circ \mathbf{A}$ are all PSD matrices.

b. $\mathcal{B} - \widetilde{\mathcal{B}} = \delta^2 (\mathcal{M} - \widetilde{\mathcal{M}})$ is also a PSD operator, which is bounded by

$$\beta \delta^2 \operatorname{tr}(\mathbf{H} \mathbf{A}) \mathbf{H} \preceq (\mathcal{B} - \widetilde{\mathcal{B}}) \circ \mathbf{A} = \delta^2 (\mathcal{M} - \widetilde{\mathcal{M}}) \circ \mathbf{A} \preceq \delta^2 \mathcal{M} \circ \mathbf{A} \preceq \psi \delta^2 \operatorname{tr}(\mathbf{H} \mathbf{A}) \mathbf{H}.$$

*Proof.* a. Let $\mathbf{A}$ denote any PSD matrix, and $\mathbf{v}$ be any vector. We then have

$$\mathbf{v}^\top (\mathcal{M} \circ \mathbf{A}) \mathbf{v} = \mathbb{E}[(\mathbf{v}^\top \mathbf{x})^2 (\mathbf{x}^\top \mathbf{A} \mathbf{x})] \geq 0,$$

where the equality holds because $(\mathbf{v}^\top \mathbf{x})^2 \geq 0$ and $\mathbf{x}^\top \mathbf{A} \mathbf{x} \geq 0$. Furthermore,

$$\mathbf{v}^\top (\mathcal{B} \circ \mathbf{A}) \mathbf{v} = \mathbb{E}[\mathbf{v}^\top (\mathbf{I} - \delta \mathbf{x} \mathbf{x}^\top) \mathbf{A} (\mathbf{I} - \delta \mathbf{H} \mathbf{x} \mathbf{x}^\top) \mathbf{v}] = \mathbb{E}[(\mathbf{v} - \delta (\mathbf{v}^\top \mathbf{x}) \mathbf{x})^\top \mathbf{A} (\mathbf{v} - \delta (\mathbf{v}^\top \mathbf{x}) \mathbf{x})] \geq 0,$$

where the inequality holds because for any vector $\mathbf{u}$ ($\mathbf{u} = \mathbf{v} - \delta (\mathbf{v}^\top \mathbf{x}) \mathbf{x}$ in this case), we have $\mathbf{u}^\top \mathbf{A} \mathbf{u} \geq 0$. Finally, $\widetilde{\mathcal{M}}$ and $\widetilde{\mathcal{B}}$ are PSD operators because any matrix similar to a PSD matrix is also a PSD matrix.

b. The difference between $\mathcal{B}$ and $\widetilde{\mathcal{B}}$ is

$$\begin{aligned}
\mathcal{B} - \widetilde{\mathcal{B}} &= \mathbb{E}[(\mathbf{I} - \delta \mathbf{x} \mathbf{x}^\top) \otimes (\mathbf{I} - \delta \mathbf{x} \mathbf{x}^\top)] - (\mathbf{I} - \delta \mathbf{H}) \otimes (\mathbf{I} - \delta \mathbf{H}) \\
&= (\mathbf{I} \otimes \mathbf{I} - \delta \mathbf{H} \otimes \mathbf{I} - \delta \mathbf{I} \otimes \mathbf{H} + \delta^2 \mathcal{M}) - (\mathbf{I} \otimes \mathbf{I} - \delta \mathbf{H} \otimes \mathbf{I} - \delta \mathbf{I} \otimes \mathbf{H} + \delta^2 \widetilde{\mathcal{M}}) \\
&= \delta^2 (\mathcal{M} - \widetilde{\mathcal{M}}).
\end{aligned}$$

Furthermore,

$$\mathcal{M} - \widetilde{\mathcal{M}} = \mathbb{E}[(\mathbf{x} \mathbf{x}^\top - \mathbf{H}) \otimes (\mathbf{x} \mathbf{x}^\top - \mathbf{H})],$$

so $\mathcal{M} - \widetilde{\mathcal{M}}$ is a PSD operator. The upper bound follows directly from the fact that $\widetilde{\mathcal{M}}$ is PSD and Assumption 3.2.

$\square$

Lemma G.2 and Lemma G.3 are similar to their counterparts in Zou et al. (2021), and are presented here for completeness.

**Lemma G.2.** The excess risk is equivalent to

$$L(\overline{\mathbf{w}}_N) - L(\mathbf{w}_*) = \frac{1}{2} \langle \mathbf{H}, \overline{\boldsymbol{\eta}}_N \otimes \overline{\boldsymbol{\eta}}_N \rangle.$$

*Proof.* By definition of the risk function, we have

$$L(\overline{\mathbf{w}}_N) - L(\overline{\mathbf{w}}_*) = \frac{1}{2} \mathbb{E}_{(\mathbf{x}, y) \sim \mathcal{D}}[(y - \langle \overline{\mathbf{w}}_N, \mathbf{x} \rangle)^2 - (y - \langle \mathbf{w}_*, \mathbf{x} \rangle)^2]$$

$$= \frac{1}{2}\mathbb{E}_{(\mathbf{x},y)\sim\mathcal{D}}[(\mathbf{w}_* - \overline{\mathbf{w}}_N)^\top(\mathbf{x}\cdot 2y - \mathbf{x}\mathbf{x}^\top(\overline{\mathbf{w}}_N + \mathbf{w}_*))]$$

$$= \frac{1}{2}(\mathbf{w}_* - \overline{\mathbf{w}}_N)^\top(2\mathbf{H}\mathbf{w}_* - \mathbf{H}(\overline{\mathbf{w}}_N + \mathbf{w}_*))$$

$$= \frac{1}{2}\langle\mathbf{H}, \overline{\boldsymbol{\eta}}_N \otimes \overline{\boldsymbol{\eta}}_N\rangle,$$

where the third equality holds due to (B.1) and the definition of $\mathbf{H}$. $\qquad\square$

**Lemma G.3.** For any $t > 0$, we have

$$\boldsymbol{\eta}_t = \boldsymbol{\eta}_t^{\mathrm{bias}} + \boldsymbol{\eta}_t^{\mathrm{var}}.$$

We thus have

$$\overline{\boldsymbol{\eta}}_N = \overline{\boldsymbol{\eta}}_N^{\mathrm{bias}} + \overline{\boldsymbol{\eta}}_N^{\mathrm{var}}.$$

*Proof.* We prove the lemma by induction. When $t = 0$, the lemma holds trivially. Suppose that the lemma holds for $t - 1$, then we have

$$\boldsymbol{\eta}_t = \mathbf{w}_t - \mathbf{w}_* = (\mathbf{w}_{t-1} - \mathbf{w}_*) + \delta(y_t - \langle\mathbf{w}_{t-1}, \mathbf{x}_t\rangle)\mathbf{x}_t$$

$$= (\mathbf{w}_{t-1} - \mathbf{w}_*) + \delta(\xi_t - \langle\mathbf{w}_{t-1} - \mathbf{w}_*, \mathbf{x}_t\rangle)\mathbf{x}_t$$

$$= (\mathbf{I} - \delta\mathbf{x}_t\mathbf{x}_t^\top)\boldsymbol{\eta}_{t-1} + \delta\xi_t\mathbf{x}_t$$

$$= (\mathbf{I} - \delta\mathbf{x}_t\mathbf{x}_t^\top)(\boldsymbol{\eta}_t^{\mathrm{bias}} + \boldsymbol{\eta}_t^{\mathrm{var}}) + \delta\xi_t\mathbf{x}_t$$

$$= [(\mathbf{I} - \delta\mathbf{x}_t\mathbf{x}_t^\top)\boldsymbol{\eta}_t^{\mathrm{bias}}] + [(\mathbf{I} - \delta\mathbf{x}_t\mathbf{x}_t^\top)\boldsymbol{\eta}_t^{\mathrm{var}} + \delta\xi_t\mathbf{x}_t]$$

$$= \boldsymbol{\eta}_t^{\mathrm{bias}} + \boldsymbol{\eta}_t^{\mathrm{var}},$$

where the fifth equality holds due to the induction hypothesis. Therefore, the lemma holds for $t$. Combining all the above, the lemma is proved for all $t \geq 0$. $\qquad\square$

**Lemma G.4.** The second moment of the residual vectors can be decomposed as

$$\mathbb{E}[\overline{\boldsymbol{\eta}}_N^{\mathrm{bias}} \otimes \overline{\boldsymbol{\eta}}_N^{\mathrm{bias}}] = (1 - \alpha)^2$$

$$\cdot \sum_{t=0}^{N-1}\left(\sum_{k=0}^{N-2-t}\alpha^{N-2-t-k}(\mathbf{I} - \delta\mathbf{H})^k\right)((\mathcal{B} - \widetilde{\mathcal{B}}) \circ \mathbf{B}_t)\left(\sum_{k=0}^{N-2-t}\alpha^{N-2-t-k}(\mathbf{I} - \delta\mathbf{H})^k\right)$$

$$+ \left(\alpha^N\mathbf{I} + (1 - \alpha)\sum_{k=0}^{N-1}\alpha^{N-1-k}(\mathbf{I} - \delta\mathbf{H})^k\right)\mathbf{B}_0\left(\alpha^N\mathbf{I} + (1 - \alpha)\sum_{k=0}^{N-1}\alpha^{N-1-k}(\mathbf{I} - \delta\mathbf{H})^k\right),$$

$$\mathbb{E}[\overline{\boldsymbol{\eta}}_N^{\mathrm{var}} \otimes \overline{\boldsymbol{\eta}}_N^{\mathrm{var}}] = (1 - \alpha)^2$$

$$\cdot \sum_{t=0}^{N-1}\left(\sum_{k=0}^{N-2-t}\alpha^{N-2-t-k}(\mathbf{I} - \delta\mathbf{H})^k\right)((\mathcal{B} - \widetilde{\mathcal{B}}) \circ \mathbf{C}_t + \delta^2\boldsymbol{\Sigma})\left(\sum_{k=0}^{N-2-t}\alpha^{N-2-t-k}(\mathbf{I} - \delta\mathbf{H})^k\right).$$

*Proof.* To simplify notations, we omit the superscripts of $\boldsymbol{\eta}_t$ and $\overline{\boldsymbol{\eta}}_N$, and denote $\mathbf{D}_t = \mathbb{E}[\boldsymbol{\eta}_t \otimes \boldsymbol{\eta}_t]$. According to the definition of $\overline{\boldsymbol{\eta}}_N$, we have

$$\mathbb{E}[\overline{\boldsymbol{\eta}}_N \otimes \overline{\boldsymbol{\eta}}_N] = \mathbb{E}\left[\left(\alpha^N\boldsymbol{\eta}_0 + (1 - \alpha)\sum_{t=0}^{N-1}\alpha^{N-1-t}\boldsymbol{\eta}_t\right) \otimes \left(\alpha^N\boldsymbol{\eta}_0 + (1 - \alpha)\sum_{t=0}^{N-1}\alpha^{N-1-t}\boldsymbol{\eta}_t\right)\right]$$

$$= \alpha^{2N}\mathbf{D}_0 + (1 - \alpha)\sum_{t=0}^{N-1}\alpha^{2N-1-t}\Big[\mathbb{E}[\boldsymbol{\eta}_0 \otimes \boldsymbol{\eta}_t] + \mathbb{E}[\boldsymbol{\eta}_t \otimes \boldsymbol{\eta}_0]\Big]$$

$$+ (1 - \alpha)^2\sum_{s=0}^{N-1}\sum_{t=0}^{N-1}\alpha^{2N-2-s-t}\mathbb{E}[\boldsymbol{\eta}_s \otimes \boldsymbol{\eta}_t]$$

$$= \alpha^{2N}\mathbf{D}_0 + (1 - \alpha)\sum_{t=0}^{N-1}\alpha^{2N-1-t}[\mathbf{D}_0(\mathbf{I} - \delta\mathbf{H})^t + (\mathbf{I} - \delta\mathbf{H})^t\mathbf{D}_0]$$

$$+ (1-\alpha)^2 \sum_{t=0}^{N-1} \left[ \alpha^{2N-2-2t} \mathbf{D}_t + \sum_{k=1}^{N-1-t} \alpha^{2N-2-2t-k} [\mathbf{D}_t(\mathbf{I}-\delta\mathbf{H})^k + (\mathbf{I}-\delta\mathbf{H})^k \mathbf{D}_t] \right]$$

$$= \left( \alpha^N \mathbf{I} + (1-\alpha) \sum_{k=0}^{N-1} \alpha^{N-1-k}(\mathbf{I}-\delta\mathbf{H})^k \right) \mathbf{D}_0 \left( \alpha^N \mathbf{I} + (1-\alpha) \sum_{k=0}^{N-1} \alpha^{N-1-k}(\mathbf{I}-\delta\mathbf{H})^k \right)$$

$$- (1-\alpha)^2 \left( \sum_{k=0}^{N-1} \alpha^{N-1-k}(\mathbf{I}-\delta\mathbf{H})^k \right) \mathbf{D}_0 \left( \sum_{k=0}^{N-1} \alpha^{N-1-k}(\mathbf{I}-\delta\mathbf{H})^k \right)$$

$$+ (1-\alpha)^2 \sum_{t=0}^{N-1} \left[ \left( \sum_{k=0}^{N-1-t} \alpha^{N-1-t-k}(\mathbf{I}-\delta\mathbf{H})^k \right) \mathbf{D}_t \left( \sum_{k=0}^{N-1-t} \alpha^{N-1-t-k}(\mathbf{I}-\delta\mathbf{H})^k \right) \right.$$

$$\left. - \left( \sum_{k=1}^{N-1-t} \alpha^{N-1-t-k}(\mathbf{I}-\delta\mathbf{H})^k \right) \mathbf{D}_t \left( \sum_{k=1}^{N-1-t} \alpha^{N-1-t-k}(\mathbf{I}-\delta\mathbf{H})^k \right) \right]$$

$$= \left( \alpha^N \mathbf{I} + (1-\alpha) \sum_{k=0}^{N-1} \alpha^{N-1-k}(\mathbf{I}-\delta\mathbf{H})^k \right) \mathbf{D}_0 \left( \alpha^N \mathbf{I} + (1-\alpha) \sum_{k=0}^{N-1} \alpha^{N-1-k}(\mathbf{I}-\delta\mathbf{H})^k \right)$$

$$+ (1-\alpha)^2 \sum_{t=0}^{N-2} \left[ \left( \sum_{k=0}^{N-2-t} \alpha^{N-2-t-k}(\mathbf{I}-\delta\mathbf{H})^k \right) \mathbf{D}_{t+1} \left( \sum_{k=0}^{N-2-t} \alpha^{N-2-t-k}(\mathbf{I}-\delta\mathbf{H})^k \right) \right.$$

$$\left. - \left( \sum_{k=1}^{N-1-t} \alpha^{N-1-t-k}(\mathbf{I}-\delta\mathbf{H})^k \right) \mathbf{D}_t \left( \sum_{k=1}^{N-1-t} \alpha^{N-1-t-k}(\mathbf{I}-\delta\mathbf{H})^k \right) \right]$$

$$= \left( \alpha^N \mathbf{I} + (1-\alpha) \sum_{k=0}^{N-1} \alpha^{N-1-k}(\mathbf{I}-\delta\mathbf{H})^k \right) \mathbf{D}_0 \left( \alpha^N \mathbf{I} + (1-\alpha) \sum_{k=0}^{N-1} \alpha^{N-1-k}(\mathbf{I}-\delta\mathbf{H})^k \right)$$

$$+ (1-\alpha)^2 \sum_{t=0}^{N-1} \left( \sum_{k=0}^{N-2-t} \alpha^{N-2-t-k}(\mathbf{I}-\delta\mathbf{H})^k \right) (\mathbf{D}_{t+1} - \widetilde{\mathcal{B}} \circ \mathbf{D}_t) \left( \sum_{k=0}^{N-2-t} \alpha^{N-2-t-k}(\mathbf{I}-\delta\mathbf{H})^k \right),$$

where the third inequality holds because $\mathbb{E}[\boldsymbol{\eta}_{t+k} \otimes \boldsymbol{\eta}_t] = \mathbb{E}[\mathbb{E}[\boldsymbol{\eta}_{t+k} \otimes \boldsymbol{\eta}_t | \mathcal{F}_t]] = \mathbb{E}[(\mathbf{I}-\delta\mathbf{H})^k(\boldsymbol{\eta}_t \otimes \boldsymbol{\eta}_t)] = (\mathbf{I}-\delta\mathbf{H})^k \mathbf{D}_t$, and the fifth equality holds due to telescope sum. Specifically, for the bias residual, we have $\mathbf{B}_{t+1} = \mathcal{B} \circ \mathbf{B}_t$, so

$$\mathbb{E}[\overline{\boldsymbol{\eta}}_N^{\text{bias}} \otimes \overline{\boldsymbol{\eta}}_N^{\text{bias}}] = (1-\alpha)^2$$

$$\cdot \sum_{t=0}^{N-1} \left( \sum_{k=0}^{N-2-t} \alpha^{N-2-t-k}(\mathbf{I}-\delta\mathbf{H})^k \right) ((\mathcal{B} - \widetilde{\mathcal{B}}) \circ \mathbf{B}_t) \left( \sum_{k=0}^{N-2-t} \alpha^{N-2-t-k}(\mathbf{I}-\delta\mathbf{H})^k \right)$$

$$+ \left( \alpha^N \mathbf{I} + (1-\alpha) \sum_{k=0}^{N-1} \alpha^{N-1-k}(\mathbf{I}-\delta\mathbf{H})^k \right) \mathbf{B}_0 \left( \alpha^N \mathbf{I} + (1-\alpha) \sum_{k=0}^{N-1} \alpha^{N-1-k}(\mathbf{I}-\delta\mathbf{H})^k \right).$$

For the variance residual, we have $\mathbf{C}_{t+1} = \mathcal{B} \circ \mathbf{C}_t + \delta^2 \boldsymbol{\Sigma}$ and $\mathbf{C}_0 = \mathbf{0}$, so

$$\mathbb{E}[\overline{\boldsymbol{\eta}}_N^{\text{var}} \otimes \overline{\boldsymbol{\eta}}_N^{\text{var}}] = (1-\alpha)^2$$

$$\cdot \sum_{t=0}^{N-1} \left( \sum_{k=0}^{N-2-t} \alpha^{N-2-t-k}(\mathbf{I}-\delta\mathbf{H})^k \right) ((\mathcal{B} - \widetilde{\mathcal{B}}) \circ \mathbf{C}_t + \delta^2 \boldsymbol{\Sigma}) \left( \sum_{k=0}^{N-2-t} \alpha^{N-2-t-k}(\mathbf{I}-\delta\mathbf{H})^k \right).$$

$\square$

## H  EXCESS RISK BOUND FOR ARBITRARY AVERAGING SCHEME

**Theorem H.1.** For SGD with any averaging scheme defined as $\overline{\mathbf{w}}_N = \beta_0 \mathbf{w}_0 + \sum_{t=0}^{N-1}(\beta_{t+1} - \beta_t)\mathbf{w}_t$, the effective bias error satisfies

$$\text{EffectiveBias} \leq \sum_{i=1}^{d} \lambda_i (\mathbf{w}_0 - \mathbf{w}_*)^2 \cdot b_i^2, \quad \text{where} \quad b_i = \beta_0 + \sum_{k=0}^{N-1}(\beta_{k+1} - \beta_k)(1-\delta\lambda_i)^k.$$

The effective variance error satisfies

$$\text{EffectiveVar} \leq \frac{\sigma^2}{1 - \psi\delta\operatorname{tr}(\mathbf{H})} \sum_{i=1}^{d}\sum_{t=1}^{N-1}(\delta\lambda_i)^2 \bigg(\sum_{k=t}^{N-1}(\beta_{k+1} - \beta_k)(1 - \delta\lambda_i)^{k-t}\bigg)^2$$

$$+ \bigg[\sum_{i=1}^{d}\Big((\delta\lambda_i) \wedge \max_k\{\beta_{k+1} - \beta_k\}\Big)^2\bigg] \cdot \frac{\psi(\|\mathbf{w}_0 - \mathbf{w}_*\|_{\mathbf{I}_{0:k^\dagger}}^2 + N\delta\|\mathbf{w}_0 - \mathbf{w}_*\|_{\mathbf{H}_{k^\dagger}}^2)}{\delta(1 - \psi\delta\operatorname{tr}(\mathbf{H}))},$$

where $k^\dagger = \max\{i : \lambda_i \geq 1/(N\delta)\}$.

*Proof.* We follow the bias-variance decomposition of Lemma C.1. The bias error satisfies

$$\langle \mathbf{H}, \mathbb{E}[\overline{\boldsymbol{\eta}}_N^{\text{bias}} \otimes \overline{\boldsymbol{\eta}}_N^{\text{bias}}]\rangle$$

$$= \bigg\langle \mathbf{H}, \bigg[\beta_0\mathbf{I} + \sum_{k=0}^{N-1}(\beta_{k+1} - \beta_k)(\mathbf{I} - \delta\mathbf{H})^k\bigg]\mathbf{B}_0\bigg[\beta_0\mathbf{I} + \sum_{k=0}^{N-1}(\beta_{k+1} - \beta_k)(\mathbf{I} - \delta\mathbf{H})^k\bigg]\bigg\rangle$$

$$+ \sum_{t=1}^{N-1}\bigg\langle \mathbf{H}, \bigg[\sum_{k=t}^{N-1}(\beta_{k+1} - \beta_k)(\mathbf{I} - \delta\mathbf{H})^{k-t}\bigg](\mathbf{B}_t - \widetilde{\mathcal{B}} \circ \mathbf{B}_{t-1})\bigg[\sum_{k=t}^{N-1}(\beta_{k+1} - \beta_k)(\mathbf{I} - \delta\mathbf{H})^{k-t}\bigg]\bigg\rangle$$

$$\preceq \underbrace{\sum_{i=1}^{d}\lambda_i(\mathbf{w}_0 - \mathbf{w}_*)_i^2 \cdot \bigg(\beta_0 + \sum_{k=0}^{N-1}(\beta_{k+1} - \beta_k)(1 - \delta\lambda_i)^k\bigg)^2}_{K_1}$$

$$+ \underbrace{\sum_{i=1}^{d}\sum_{t=1}^{N-1}(\delta\lambda_i)^2\operatorname{tr}(\mathbf{H}\mathbf{B}_{t-1})\bigg(\sum_{k=t}^{N-1}(\beta_{k+1} - \beta_k)(1 - \delta\lambda_i)^{k-t}\bigg)^2}_{K_2}$$

where the second inequality used the property $\mathbf{B}_t - \widetilde{\mathcal{B}} \circ \mathbf{B}_{t-1} = (\mathcal{B} - \widetilde{\mathcal{B}}) \circ \mathbf{B}_{t-1} \preceq \psi\delta^2\operatorname{tr}(\mathbf{H}\mathbf{B}_{t-1})\mathbf{H}$. The term $K_1$ is the effective bias, and is equal to $\sum_i \lambda_i(\mathbf{w}_0 - \mathbf{w}_*)_i^2 \cdot b_i^2$. For the term $K_2$, we firstly have

$$(\delta\lambda_i)^2\operatorname{tr}(\mathbf{H}\mathbf{B}_{t-1})\bigg(\sum_{k=t}^{N-1}(\beta_{k+1} - \beta_k)(1 - \delta\lambda_i)^{k-t}\bigg)^2 \leq \Big(\max_k\{\beta_{k+1} - \beta_k\}\Big)^2\sum_{t=1}^{N-1}\operatorname{tr}(\mathbf{H}\mathbf{B}_{t-1}),$$

where the inequality holds because $\sum_{k=t}^{N-1}(\beta_{k+1} - \beta_k)(1 - \delta\lambda_i)^{k-t} \leq \max_k\{\beta_{k+1} - \beta_k\}\sum_{k=t}^{N-1}(1 - \delta\lambda_i)^{k-t} \leq \max_k\{\beta_{k+1} - \beta_k\}(\delta\lambda_i)^{-1}$. We also have

$$(\delta\lambda_i)^2\operatorname{tr}(\mathbf{H}\mathbf{B}_{t-1})\bigg(\sum_{k=t}^{N-1}(\beta_{k+1} - \beta_k)(1 - \delta\lambda_i)^{k-t}\bigg)^2 \leq (\delta\lambda_i)^2\sum_{i=1}^{N-1}\operatorname{tr}(\mathbf{H}\mathbf{B}_{t-1}),$$

where the inequality holds because $\sum_{k=t}^{N-1}(\beta_{k+1} - \beta_k)(1 - \delta\lambda_i)^{k-t} \leq \sum_{k=t}^{N-1}(\beta_{k+1} - \beta_k) = \beta_N - \beta_t \leq \beta_N = 1$. Combining with Lemma E.2, we have

$$K_2 \leq \bigg[\sum_{i=1}^{d}\Big((\delta\lambda_i) \wedge \max_k\{\beta_{k+1} - \beta_k\}\Big)^2\bigg] \cdot \frac{\psi(\|\mathbf{w}_0 - \mathbf{w}_*\|_{\mathbf{I}_{0:k^\dagger}}^2 + N\delta\|\mathbf{w}_0 - \mathbf{w}_*\|_{\mathbf{H}_{k^\dagger}}^2)}{\delta(1 - \psi\delta\operatorname{tr}(\mathbf{H}))}.$$

For the variance error, we have

$$\langle \mathbf{H}, \mathbb{E}[\overline{\boldsymbol{\eta}}_N^{\text{var}} \otimes \overline{\boldsymbol{\eta}}_N^{\text{var}}]\rangle$$

$$= \sum_{t=1}^{N-1}\bigg\langle \mathbf{H}, \bigg[\sum_{k=t}^{N-1}(\beta_{k+1} - \beta_k)(\mathbf{I} - \delta\mathbf{H})^{k-t}\bigg](\mathbf{C}_t - \widetilde{\mathcal{B}} \circ \mathbf{C}_{t-1})\bigg[\sum_{k=t}^{N-1}(\beta_{k+1} - \beta_k)(\mathbf{I} - \delta\mathbf{H})^{k-t}\bigg]\bigg\rangle$$

$$\leq \frac{\sigma^2}{1 - \psi\delta\operatorname{tr}(\mathbf{H})}\sum_{i=1}^{d}\sum_{t=1}^{N-1}(\delta\lambda_i)^2\bigg(\sum_{k=t}^{N-1}(\beta_{k+1} - \beta_k)(1 - \delta\lambda_i)^{k-t}\bigg)^2.$$

where we follow the proof of Lemma C.2 and use $\mathbf{C}_t - \widetilde{\mathcal{B}} \circ \mathbf{C}_{t-1} \preceq \frac{\sigma^2\delta^2}{1 - \psi\delta\operatorname{tr}(\mathbf{H})}\mathbf{H}$. $\qquad\square$

