# OpenReview forum: "Understanding SGD with Exponential Moving Average: A Case Study in Linear Regression"
_ICLR.cc/2026/Conference — Submitted to ICLR 2026_

### Official Review · Reviewer_YCSc · 2025-10-27

**Soundness:** 3
**Presentation:** 2
**Contribution:** 3
**Rating:** 6
**Confidence:** 2

**Summary:**

This paper focuses on methods for linear regression that run stochastic gradient descent (SGD) while maintaining an exponential moving average (EMA), i.e., a smoothed version of the SGD iterates. The main results provide upper and lower bounds on the bias and variance of the smoothed weight vector.

The derived bounds depend on:
- the learning rate for SGD,
- the averaging parameter,
- the spectrum of the feature covariance matrix,
- bounds on the fourth moments of the feature matrix, and
- a decoupling assumption between the label noise and the features.

Overall, this is a reasonably clear paper that provides some novel theoretical results for a specific method. There seems to be growing interest in these types of methods within the community, so the topic may be of sufficient relevance. One could argue that the specific results are somewhat incremental given recent work on averaging schemes, but the key innovations in the proof technique appear to extend more generally to any time-dependent averaging scheme.

There is room for improvement in the presentation. In particular, the notation is quite heavy, and I found the discussion of the results difficult to follow at times.


Suggestions for improvement:
- The discussion in Section 4.2 about interpreting the different terms in the variance was difficult to follow. Breaking the argument into separate parts or labeling the individual terms more explicitly would help the reader understand the contribution of each component.-
- Proposition 4.3 was also challenging to parse. It would help to provide more intuition for the different parameter regimes (sample size, learning rate, averaging rate) and to clarify their practical meaning. The red highlighting was not helpful for me.
- It would be valuable to include some guidance for practice: How should one choose the parameters in realistic scenarios? Is there a way to adjust the level of averaging adaptively during training?
- It would also improve readability if the colors were matched between subfigures (a) and (b) in Figure 1.

**Strengths:**

- The explicit bounds on the bias and variance is an interesting and novel result.  It extends previous work on iterate and tail averaging to a more general, time-dependent averaging framework.

- The assumptions are clearly stated and are quite general. .

- While the basic setup of the proof follows closely from prior work, the main technical novelty seems to be the ability to handle the general time-dependent averaging scheme described in Section 7, which includes many variants of SGD as special cases.

- The outline of the proof in Section 7 is helpful.

**Weaknesses:**

-   The presentation is dense and notation-heavy, making several sections difficult to follow.

- The paper could include more practical guidance — e.g., how to select the averaging parameter alpha based on the observed data.

- It is not clear to me how much this method improves upon other averaging schemes, so the overall impact may be limited.  For example, analysis of non-linear models would likely have broader impact.

**Questions:**

- Would it be possible to state a prove  a general theoretical result for any generic averaging scheme? What really matters in the weights to ensure good performance? Can the averaging be data-dependent?

- My interpretation of Table 1 and the figures is that all reasonable averaging schemes are somewhat equivalent. It this a reasonable takeaway or is there really some special about exponential moving average that I am missing?

---

> ### Author Response · Authors · 2025-11-20
>
> Many thanks for the positive review!
>
> **Q1 (Presentation).** The presentation is dense and notation-heavy, making several sections difficult to follow.
>
> The discussion in Section 4.2 about interpreting the different terms in the variance was difficult to follow. Breaking the argument into separate parts or labeling the individual terms more explicitly would help the reader understand the contribution of each component.
>
> Proposition 4.3 was also challenging to parse. It would help to provide more intuition for the different parameter regimes (sample size, learning rate, averaging rate) and to clarify their practical meaning. The red highlighting was not helpful for me.
>
> **A1.** We apologize for the confusion, and appreciate the reviewer for the valuable suggestion.
>
> - For the discussion of the effective variance error, we have add labels in Theorem 4.1 to discuss the terms in greater detail. We have also added discussions of the terms.
> - For Proposition 4.3, our purpose is to discuss the order of $b_i$, especially the exponential decay rate (highlighted in red) in different regimes of the **eigenvalue subspace** rather than the parameter regime. We have adjusted the wording for better understanding. We have also added an illustrating heatmap of $b_i$ with different regimes.
>
> Please refer to the revised paper for details. We would be happy to provide new revisions with further suggestions from the reviewer.
>
> ---
>
> **Q2 (Choice of $\alpha$).** It would be valuable to include some guidance for practice: How should one choose the parameters in realistic scenarios? Is there a way to adjust the level of averaging adaptively during training?
>
> **A2.** We have added a heuristical discussion on the choice of $\alpha$ at the end of Section 4. However, adjusting $\alpha$ during training is not common practice in training modern deep learning models.
>
> ---
>
> **Q3 (Color in Figure 1).** It would also improve readability if the colors were matched between subfigures (a) and (b) in Figure 1.
>
> **A3.** Many thanks for the suggestion. The confusion of Figure 1 is mainly due to the an issue of the legend; the colors for the same averaging scheme of the two subfigures are the same. We have fixed the issue in the revision.
>
> ---
>
> **Q4 (Broader impacts).** It is not clear to me how much this method improves upon other averaging schemes, so the overall impact may be limited. For example, analysis of non-linear models would likely have broader impact.
>
> **A4.** Extending our analysis to models that are more similar to real-world models is an exciting and promising direction. For example, we can combine our analysis and the analysis in [1] to study single-index models, i.e., single-neuron network with ReLU activation. Please refer to Q/A8 for a detailed comparison between EMA and other averaging schemes.
>
> ---
>
> [1] Wu et al. Finite-sample analysis of learning high-dimensional single relu neuron. 2023.

---

> > ### Comment · Reviewer_YCSc · 2025-11-22
> >
> > I appreciate your responses to my questions. I provide some additional feedback and suggestions below.
> >
> > Q1. I think there is still  room for improvement in the presentation of Proposition 4.3. A clearer structure would help highlight the key scaling properties across the four regimes, which are among the main takeaways of your detailed analysis.
> > Consider using a cases environment or a table to eliminate redundant text and make it easier to directly compare both the scalings and the intervals defining the four cases. You could also give the four regimes short descriptive names to help the reader.
> >
> > Q3. I appreciate the updated colors and figures. However, it remains somewhat difficult to interpret Figure 2 due to the overlapping shading and the similarity among color types. Enhancing the contrast or using more distinguishable patterns would make the figure clearer.

---

> ### Author Response · Authors · 2025-11-20
>
> **Q5.** Would it be possible to state a prove a general theoretical result for any generic averaging scheme?
>
> **A5.** We appreciate the reviewer for the proposal. It is possible to prove a general theoretical result for any averaging scheme. We have included the results in Appendix H in the revised manuscript.
>
> ---
>
> **Q6.** What really matters in the weights to ensure good performance?
>
> **A6.** The bias-variance tradeoff persists in almost every averaging scheme, and a good choice of averaging parameters should achieve a balance between noise suppression and bias contraction. For example, in EMA, increasing $\alpha$ decreases the effective variance error smaller but makes the decay rate of the effective bias error slower. The optimal choice of $\alpha$ achieved at the minimum total excess risk
>
> We would like to kindly ask for the clarification of this question from the reviewer. Does the *weight* here mean the hyperparameter $\alpha$ in EMA, or does it stand for the weights for an arbitrary averaging scheme?
>
> ---
>
> **Q7.** Can the averaging be data-dependent?
>
> **A7.** Data-dependent averaging is possible, but not commonly used in training modern deep learning models. Boosting algorithms can generate data-dependent weights to perform model averaging, e.g., in AdaBoost [2]. However, we do not discuss these methods because they are rarely used in the practice of training modern deep learning models.
>
> ---
>
> **Q8.** My interpretation of Table 1 and the figures is that all reasonable averaging schemes are somewhat equivalent. It this a reasonable takeaway or is there really some special about exponential moving average that I am missing?
>
> **A8.** We would like to correct the misunderstanding of the reviewer: The performances of different averaging schemes are very different.
>
> - **Bias error.** We prove that the effective bias error of SGD with EMA decays exponentially **in the total number of iterations** $N$, a property shared only by SGD without averaging (in Theorem 4.2). This phenomenon is observed in the experiments: In Figure 1(a), the bias error of SGD with EMA goes below all other averaging schemes.
> - **Variance error.** We prove that the effective variance error of SGD with EMA **increases** as $N$ increases, which is different from iterate averaging and tail averaging. This phenomenon is clearly illustrated in Figure 1(b). Despite this, the effective variance error is still moderate compared with other averaging schemes.
> - **Effective dimension.** The effective dimension of SGD with EMA is $\max_i\{\lambda_i\ge(1-\alpha)/\delta\}$, which is irrelevant to $N$. Therefore, unlike other averaging schemes, the effective dimension does not go to the ambient dimension $d$ when $N\to\infty$.
>
> We would apologize again for the misinterpretation caused by the legend of Figure 1(b).
>
> [2] Freund and Shapire. A decision-theoretic generalization of on-line learning and an application to boosting. 1997.

---

> > ### Comment · Reviewer_YCSc · 2025-11-22
> >
> > Q5. It’s nice to see the extended analysis for general averaging schemes included in Appendix H.
> >
> > Q6. Q8. My earlier comments about weights referred to general weighting schemes, not just the EMA parameter $\alpha$. To provide a bit more clarity, I understand that for a given problem setting, the bias–variance pair depends on the parameters $(\alpha, N)$ for EMA and on the parameters $(s, N)$ for tail averaging.
> >
> > On one hand, as you note in your response to Q8, for fixed $\alpha$ and $s$, the scaling with respect to $N$ differs substantially. On the other hand, if one optimizes over $\alpha$ or $s$ for a given $N$, then Figure 2 suggests that the performance of the two methods becomes roughly comparable. In other words, for fixed $N$, the bias–variance trade-off achievable by varying $\alpha$ in EMA appears similar to that obtained by varying $s$ in tail averaging.
> >
> > After revisiting your paper, I see that you do touch on this in lines 72–73, noting that matching the same scaling with tail averaging requires adjusting the tail parameter according to the time horizon $N$. This addresses my earlier question about what makes EMA special—namely, that its bias–variance trade-off remains stable with respect to $N$. It might help to reiterate this point explicitly when discussing your empirical results.

---

> > > ### Author Response · Authors · 2025-11-23
> > >
> > > Thank you very much for the followup discussions!
> > >
> > > **Q1.** We appreciate the reviewer for the detailed suggestions. We have modified Proposition 4.3 using the `cases` environment. We also added Table 2 to illustrate the four regimes, with an interpretation of the condition for each regime.
> > >
> > > **Q3.** In the latest manuscript, we have removed the confidence bands in Figure 2, and have deferred the original Figure 2 with confidence bands to the appendix. We hope this can make the interpretations of the figure easier. The different shades of blue in Figure 2 stand for tail averaging with different choices of $s$, and we hope the similarity of colors causes less confusion without the confidence bands.
> > >
> > > **Q6/8.** We appreciate the reviewer for the recognition of our finding that the variance error of SGD with EMA is similar to SGD with tail averaging when $(1-\alpha)(N-s)\simeq1$. We have also added the interpretation of Figure 2(b) concerning this phenomenon, by observing the intersecting point of the variance curves of SGD with EMA and tail averaging.

---

### Official Review · Reviewer_867M · 2025-10-29

**Soundness:** 3
**Presentation:** 3
**Contribution:** 3
**Rating:** 4
**Confidence:** 4

**Summary:**

This paper  investigates the generalization behavior of constant step-size Stochastic Gradient Descent (SGD) when combined with Exponential Moving Average (EMA), using high-dimensional linear regression as a prototypical setting. The authors derive instance-dependent and dimension-independent upper and lower bounds for the excess risk, which are tied to the spectral properties of the data covariance matrix. The analysis provides a clear decomposition of the error into an "effective bias'' and an "effective variance.'' The effective bias is shown to decay exponentially in each feature subspace. The effective variance is demonstrated to be jointly controlled by the EMA parameter $\alpha$, the learning rate $\delta$, the number of iterations $N$, and the covariance spectrum. A key theoretical contribution is the direct comparison of EMA with tail averaging, revealing that under appropriate conditions, their variance terms are comparable and can even be equivalent in magnitude. The work also extends the analysis to the mini-batch SGD setting and discusses the resulting critical batch size.

**Strengths:**

- This paper provides the first fine-grained, instance-dependent risk bounds for SGD with EMA in the high-dimensional linear regression regime. The spectral decomposition analysis, which demonstrates that the bias decays exponentially along each eigendirection, is a solid theoretical contribution. The resulting dimension-free upper and lower bounds, characterized by two effective dimension thresholds, are technically sound and insightful.
- A significant strength is the unified framework used to compare EMA against classical strategies like no averaging, full-iterate averaging, and tail averaging. The analysis lucidly establishes a formal connection between EMA and tail averaging, particularly their equivalence in variance magnitude under a specific condition. This helps clarify theoretically why EMA is often observed to be a superior or highly competitive averaging strategy in practice.

**Weaknesses:**

- The primary weakness lies in the strength of Assumption 3.3, the fourth-moment lower bound. This assumption ($\mathcal{M}\circ A - HAH \succeq \beta \mathrm{tr}(HA) H$) effectively requires a uniform "energy'' lower bound from the fourth-order moments in all directions. This is a very strong condition that is unlikely to hold for many practical high-dimensional data distributions, especially those that are non-isotropic, heavy-tailed, or mixtures, where tail subspaces might have weak energy or significant variations in directional kurtosis. This assumption appears to be a technical convenience for achieving a tight lower bound rather than a verifiable data property, which limits the robustness and generalizability of the conclusions.
- There is a notable disconnect between the paper's stated motivation and its actual contributions. The abstract and introduction heavily motivate the work by citing the widespread use of EMA in training diffusion models. However, the theoretical analysis and experiments are confined entirely to linear regression (with a brief extension to a single ReLU neuron in the appendix). The paper offers no specific analysis or empirical evidence related to the unique loss landscapes, model architectures, or training dynamics of diffusion models, failing to fully bridge the gap between the simple theoretical setting and the complex, practical application it claims to address.
- The experimental validation feels incomplete and offers limited practical guidance. The authors select the learning rate $\delta$ based on the derived theoretical threshold  but do not perform any systematic sweep or ablation study over this hyperparameter. Without comparing performance across a range of learning rates, it is impossible to assess the necessity, conservativeness, or near-optimality of this theoretical threshold in practice, which diminishes the practical value of the theoretical findings.

**Questions:**

Could the authors please comment on the realism of Assumption 3.3 and the implications of relaxing it, even if it impacts the tightness of the lower bound? Additionally, to bridge the gap between motivation and scope, I suggest either revising the abstract's focus on diffusion models to better match the linear regression results, or adding a discussion on how these insights might qualitatively extend to such complex models. Finally, to strengthen the empirical validation, would it be possible to add a learning rate ablation study for $\delta$ to test the necessity and conservativeness of your theoretical threshold? I am willing to raise my score if these concerns are satisfactorily addressed.

---

> ### Author Response · Authors · 2025-11-20
>
> Thank you very much for the feedback and the valuable suggestions!
>
> ---
>
> **Q1 (Fourth moment assumptions).** The primary weakness lies in the strength of Assumption 3.3, the fourth-moment lower bound. This assumption effectively requires a uniform "energy'' lower bound from the fourth-order moments in all directions. This is a very strong condition that is unlikely to hold for many practical high-dimensional data distributions, especially those that are non-isotropic, heavy-tailed, or mixtures, where tail subspaces might have weak energy or significant variations in directional kurtosis. This assumption appears to be a technical convenience for achieving a tight lower bound rather than a verifiable data property, which limits the robustness and generalizability of the conclusions. Could the authors please comment on the realism of Assumption 3.3 and the implications of relaxing it, even if it impacts the tightness of the lower bound?
>
> **A1.** The fourth moment lower bound is widely used in the literature, e.g., [1]. We follow the assumption to conduct a compare analysis of the lower bound. We would like to argue that the interpretation of Theorem 4.3 does not change without Assumption 3.3, and the assumptions is probably not that strong.
>
> We note that $\mathbb{E}[\mathbf{xx}^\top\mathbf{Axx}^\top]\succeq\mathbf{HAH}$ always holds. This indicates that removing Assumption 3.3 is equivalent to setting $\beta=0$. This makes the lower bound less tight because the term $\beta||\mathbf{w}\_0-\mathbf{w}\_\ast||\_{\mathbf{H}\_{k^\dagger:\infty}}^2$  disappears. However, setting $\beta=0$ does not affect the interpretation of the results. In fact, the $\beta||\mathbf{w}\_0-\mathbf{w}\_\ast||\_{\mathbf{H}\_{k^\dagger:\infty}}^2$ term is exactly where the gap between the upper and lower bounds occurs, and removing this term only makes the gap a bit wider.
>
> We would also like to argue that Assumption 3.3 is not that strong. The data distribution can be very anisotropic, with ill-posed covariance matrix $\mathbf{H}$. It is true that heavy-tailed data may not satisfy this assumption, but in this case the training often requires additional techniques, e.g., gradient clipping, which is out of the scope of this work. It is also possible to relax the assumption to the case where $\mathbf{A}$ is confined to all PSD matrices that are commutable with $\mathbf{H}$, according to [1].
>
> ---
>
> [1] Zou et al. Benign overfitting of constant-stepsize sgd for linear regression. 2021.

---

> ### Author Response · Authors · 2025-11-20
>
> **Q2 (Implication of complex models).** There is a notable disconnect between the paper's stated motivation and its actual contributions. The abstract and introduction heavily motivate the work by citing the widespread use of EMA in training diffusion models. However, the theoretical analysis and experiments are confined entirely to linear regression (with a brief extension to a single ReLU neuron in the appendix). The paper offers no specific analysis or empirical evidence related to the unique loss landscapes, model architectures, or training dynamics of diffusion models, failing to fully bridge the gap between the simple theoretical setting and the complex, practical application it claims to address. To bridge the gap between motivation and scope, I suggest either revising the abstract's focus on diffusion models to better match the linear regression results, or adding a discussion on how these insights might qualitatively extend to such complex models.
>
> **A2.** We apologize for the confusion caused by the motivation from the application of EMA in diffusion models in the abstract and the introduction. As indicated by the title, our goal is to conduct a **case study** of averaging schemes on linear regression. The statement of EMA in training diffusion models only functions as the evidence for the popularity of EMA.
>
> In the revised manuscript, we have avoided mentioning EMA as a motivating example, and the discussion of the application of EMA has been deferred to the section of related work. It would also be an interesting future direction to study how EMA affects the training of diffusion models.
>
> ---
>
>
> **Q3 (Ablation on LR).** The experimental validation feels incomplete and offers limited practical guidance. The authors select the learning rate based on the derived theoretical threshold but do not perform any systematic sweep or ablation study over this hyperparameter. Without comparing performance across a range of learning rates, it is impossible to assess the necessity, conservativeness, or near-optimality of this theoretical threshold in practice, which diminishes the practical value of the theoretical findings. To strengthen the empirical validation, would it be possible to add a learning rate ablation study to test the necessity and conservativeness of your theoretical threshold?
>
>
> **A3.** We appreciate the reviewer for the valuable suggestion. We have added the ablation studies on the learning rate in Appendix A.2 and verified our theoretical findings. Specifically, we show that the variance error blows up between $\delta=0.6$ and $0.7$ which justifies the necessity of the requirement of $\delta\le1/(\psi\mathrm{tr}(\mathbf{H}))$. Please refer to the revised manuscript for details.

---

> > ### Comment · Reviewer_867M · 2025-11-21
> >
> > Thank you for the author's reply. I already had a positive impression of the paper regarding its soundness, presentation, and contribution, and I appreciate the author's addition of experimental information about the learning rate $\delta$ during the discussion. Given that this is a theoretical work, and after the author clarified my understanding of Assumption 3.3, I believe its theoretical contribution meets the ICLR acceptance criteria. Therefore, I will adjust my rating accordingly.

---

> > > ### Author Response · Authors · 2025-11-21
> > >
> > > Thank you very much for the reply and the recognition of our work! We would also be happy to further improve the manuscript.

---

### Official Review · Reviewer_QSsy · 2025-10-30

**Soundness:** 3
**Presentation:** 3
**Contribution:** 3
**Rating:** 6
**Confidence:** 2

**Summary:**

The paper theoretically studies why EMA works with SGD. Specifically, it provides both upper and lower bound for the excess risk of EMA through variance and bias decomposition. Experiments are also conducted to show the variance and bias error of EMA to verify the theoretical claims.

**Strengths:**

1. The paper is the first to theoretically bound the excess risk of SGD with EMA despite the prevalance of this method.
2. There're some techinical novelty as pointed out by the authors from line 385-393, that previous analysis doesn't work due to the inhomogeneous $\beta_t-\beta_{t+1}$.
3. The experimental results seem to align with proposition 4.3 well.

**Weaknesses:**

1. The setting of linear regression is perhaps a bit simple given EMA is used in much harder optimization problems.
2. The proofs of Theorem 4.1 and 4.2 seem to largely follow the previous analysis, especially [1], and I'm not sure how to evaluate the contribution of this proof given I'm unfamiliar with this area and the related work.

[1] Zou, D., Wu, J., Braverman, V., Gu, Q., & Kakade, S. M. (2023). Benign overfitting of constant-stepsize SGD for linear regression. Journal of Machine Learning Research, 24(326), 1-58.

**Questions:**

1. Would it be possible to extend linear regression to more complicated (convex) functions, or more specifically the loss function in diffusion models?

---

> ### Author Response · Authors · 2025-11-20
>
> We appreciate the reviewer for the positive feedback!
>
> ---
>
> **Q1.** The setting of linear regression is perhaps a bit simple given EMA is used in much harder optimization problems. Would it be possible to extend linear regression to more complicated (convex) functions, or more specifically the loss function in diffusion models?
>
> **A1 (Extension to more complex models).** Extending our analysis to models that are more similar to real-world models is an exciting and promising direction. For example, we can combine our analysis and the analysis in [1] to study single-index models, i.e., single-neuron network with ReLU activation.
>
> ---
>
> **Q2 (Proof techniques).** The proofs of Theorem 4.1 and 4.2 seem to largely follow the previous analysis. I'm not sure how to evaluate the contribution of this proof given I'm unfamiliar with this area and the related work.
>
> **A2.** It is true that we follow the bias-variance decomposition and the analysis of the $\mathbf{B}_t$ and $\mathbf{C}_t$ in existing works. However, our treatment of the averaging schemes is innovative. Specifically, we develop a framework for studying a broad class of averaging schemes (Section 7). We have also added a general regret upper bound for in Appendix H. Please refer to the revised manuscript for details.
>
> ---
>
> [1] Wu et al. Finite-sample analysis of learning high-dimensional single relu neuron. 2023.

---

> > ### Comment · Reviewer_QSsy · 2025-11-25
> >
> > I thank the authors for the feedback. I do understand that the technique of previous work can't be applied directly due to the inhomogeneous difference of $\beta_t$, and this may be the key technical contribution. Overall, I do think the paper is well written with sound presentation, and I'll maintain my positive review of this paper.

---

> > > ### Author Response · Authors · 2025-11-25
> > >
> > > We appreciate your response, and thank you again for the positive feedback! We would be happy to further improve our manuscript if you have other suggestions.

---

### Official Review · Reviewer_Fvsd · 2025-11-01

**Soundness:** 3
**Presentation:** 3
**Contribution:** 3
**Rating:** 6
**Confidence:** 3

**Summary:**

This work studies the SGD algorithm (also extended to minibatch SGD) with EMA under the high-dimensional linear regression, which can be viewed as  a simplified model of overparameterized neural networks. This work provides a unified analytical framework for different averaging methods and shows that EMA is better on the bias error and variance error at the same time. The simulation experiments also support their results.

**Strengths:**

Strengths:

* From the theoretical perspective, this work clearly shows the advantage of the EMA averaging method compared with other methods.

**Weaknesses:**

1. As we know that high-dimensional linear regression is an important task, would it be possible to conduct experiments in a real-world setting and show that EMA has a better performance?

**Questions:**

Question:

Q1: Could this work discuss the theoretical or technical challenge of Adam with EMA intuitively?

Q2: I understand that these high-order assumptions have been used in previous works. However, it is also helpful to discuss these assumptions.

Minor Comments:

It would be better to add the theoretical line in Figure 1 to do a better comparison.

---

> ### Author Response · Authors · 2025-11-20
>
> Many thanks for the positive feedback!
>
> ---
>
> **Q1 (Real-world experiments).** As we know that high-dimensional linear regression is an important task, would it be possible to conduct experiments in a real-world setting and show that EMA has a better performance?
>
> **A1.** We thank the reviewer for the suggestion. We have included the experiments on linear regression in Section 8, and experiments on the single index model (i.e., $y=\mathrm{ReLU}(\langle\mathbf{x}, \mathbf{w}\rangle)+\epsilon$), which is closer to neural networks, in Appendix A. We have also discussed existing works [1, 2] that showed the advantage of EMA in a real-world setting in Section 2. Therefore, we find it unnecessary to conduct experiments in the real-world setting, and the experiments on synthetic data serves the purpose of verifying our theoretical findings.
>
> ---
>
> **Q2.** Discussion of higher-order assumptions.
>
> **A2.** We appreciate the reviewers for the suggestion. Following assumptions 3.2 and 3.3, we will discuss their relationship with the sub-Gaussian tail. Please refer to the revised manuscript for details.
>
> ---
>
> **Q3 (Adam with EMA).** Could this work discuss the theoretical or technical challenge of Adam with EMA intuitively?
>
> **A3.** The analysis of SGD relies on the bias-variance decomposition (Lines 356-368; Lemma C.1) which separates label noise from the iterates. However, the original bias-variance decomposition does not hold due to the nonlinearity of the Adam updates.
>
> We noticed that a recent work [3] studied the excess risk of SignSGD, an optimizer closely related to Adam, in linear regression. We conjecture that it is possible to extend the analysis of [3] to EMA, and it can be a promising future direction.
>
> ---
>
> **Q4.** It would be better to add the theoretical line in Figure 1 to do a better comparison.
>
> **A4.** Many thanks for the suggestion! We have tried to add the theoretical line, but it does not help illustration because the excess risk bounds in Theorems 4.2 and 4.3 are tight only up to constants. Therefore, the theoretical risk curve of the upper bound lies well above the actual curve.
>
> [1] Tarvainen and Valpola. Mean teachers are better role models: Weight-averaged consistency targets improve semi-supervised deep learning results. 2017.
>
> [2] Izmalov et al. Averaging weights leads to wider optima and better generalization. 2018.
>
> [3] Xiao et al. Exact Risk Curves of signSGD in High-Dimensions: Quantifying Preconditioning and Noise-Compression Effects. 2024.

---

### Author Response · Authors · 2025-12-02

Dear new area chair,

Thank you very much for taking over our submission! We would like to support your evaluation process amid the special situation.

We were fortunate to discuss our manuscript with the reviewers and received valuable suggestions. We believe that we have addressed all concerns of the reviewers:

- Reviewer 867M initially recommended borderline reject, and expressed concerns about the fourth moment lower bound assumption (Assumption 3.2). They also suggested ablation studies on the learning rate. We have fully resolved their concern by explaining that the assumption is not that strong as the reviewer suggested and that the interpretation of the theoretical results does not change without the assumption. We have also updated the manuscript with the ablation studies on the learning rate in Appendix A.2. The reviewer had increased the rating from 4 to 6 before the rollback.
- Reviewer YCSc provided a lot of suggestions on presentation. We improved the presentation based on their suggestions: In Section 4.2, we interpreted each term in the effective variance error in great detail. In Section 4.3, we added Table 2 to interpret different regimes of the eigen-subpsaces with different exponential decay rates.
- Other reviewers questioned our technical contributions. We argue that our theoretical framework can be applied to a broad range of averaging schemes (Section 7). We also added Appendix H to prove the excess risk bound for general averaging schemes.

We have implemented all the suggestions in the revised manuscript. We would really appreciate it if these updates could be considered in the evaluation.

---

### Meta-Review · Area_Chair_9QSj · 2026-01-02

**Summary:**

This work looks at the well-known SGD algorithm and propose to output the exponential moving average (EMA) of the iterations. The authors conduct convergence analysis on the simple linear regression problem. This goes against the main motivation that EMA is used in training generative models and diffusion models.

Looking at Table 1, it seems the only advantage of EMA compared to no averaging is that the variance error can be always be made less than one ($1-\alpha$) with EMA. This doesn't seem to be a significant improvement compared to the vanilla SGD (which is as expected since it is just a simple modification to SGD).

**Reviewer Concerns:**

The main concern is that the convergence analysis is conducted on the simple linear regression case. It is not close to training generative models or diffusion models, no matter how the authors spin it.

**Reviewer Scores:**

The reviewers scores seem inflated.

---

### Decision · Program_Chairs · 2026-01-26

Reject